# Impacts of climate change on the distribution pattern of Himalayan Rhubarb (*Rheum australe* D. Don) in Nepal Himalaya

Babu Ram Paudel[1,2]*, Chandra Kanta Subedi[1], Suresh Kumar Ghimire[3], Dipesh Pyakurel[4], Meena Rajbhandari[1], Ram Prasad Chaudhary[1]

1 Research Centre for Applied Science and Technology, Tribhuvan University, Kirtipur, Kathmandu, Nepal, 2 Department of Botany, Amrit Campus, Tribhuvan University, Lainchour, Kathmandu, Nepal, 3 Central Department of Botany, Tribhuvan University, Kirtipur, Kathmandu, Nepal, 4 Asia Network for Sustainable Agriculture and Bio-resources, Kathmandu, Nepal

* brp2033@gmail.com

## Abstract

While there has been substantial consensus that climate change poses a severe threat to the Himalayan biota; we still lack comprehensive data on the potential impact of climate change on the important Himalayan medicinal plants connected with the livelihood of local people and the national economy. In this study, using MaxEnt, we modelled the distribution pattern of *Rheum australe*, a medicinal plant prioritized by the Government of Nepal for the nation's economic prosperity, for the current as well as all four future projections (SSP126, SSP245, SSP370, and SSP585) to all the data available periods (2021–2040, 2041–2060, 2061–2080, and 2081–2100). In addition, we performed spatial overlay analysis to identify the optimum zones that could be developed as potential planting/conservation sites for *R. australe* in any of the projected future climate scenarios. Our results revealed that the suitability area of *R. australe* is expected to contract in all the scenarios and periods with a significant loss expected to occur in SSP585 for 2081–2100. On the individual district level, northwestern districts are expected to have a huge loss of suitable habitats in the future, while four districts: Jumla, Kalikot, Dolpa, and Jajarkot of the Karnali Province are expected to gain suitable habitats remarkably. Therefore, we suggest that the forests and rangelands of the four districts at an elevation range of 3300 m – 4,400 m could be developed as potential planting areas for commercial cultivation/for establishing genetic resource conservation centres. This finding thus has wider policy implications for both government and conservation organizations.

## Introduction

Since the industrial revolution, atmospheric carbon dioxide concentration has been increasing gradually which in turn causes the steady increase of mean annual global

**Data availability statement:** All relevant data are within the paper and its Supporting Information files.

**Funding:** The study was supported by an innovative research grant (Grant Number: TU-NPAR-078/79-4-04) from the Research Directorate, Office of the Rector, Tribhuvan University, Nepal. The funders had no role in study design, data collection and analysis, the decision to publish, or preparation of the manuscript.

**Competing interests:** The authors declare no conflicts of interest

temperatures. It is projected that by the end of the 21st century, the atmospheric carbon dioxide concentration will reach between 500–1000 ppm (part per million) and the global temperatures will rise by another 1–3.7°C [1,2]. The rise of carbon dioxide and temperature is likely to have diverse effects on global climates such as alternation in precipitation patterns, storm severity, glacial melting, and changes in sea level [3]. The synergistic effects of the rise of carbon dioxide concentration and temperature together with the climate changes are predicted to have severe impacts on Earth, ranging from individual species to entire ecosystems and from local to global scales [4]. Therefore, similar to other components of the earth's ecosystem, plants are likely to be severely affected by these anthropogenic alternations. Indeed, the increased carbon dioxide concentrations alter many vital physiological processes of the plants such as photosynthesis, respiration, and photorespiration which would indirectly affect the plant distribution [5,6]. Likewise, increased global temperature and alternation in precipitation patterns tend to alter the habitat and distribution range of plant species [7]. Studies have suggested that the rise in global temperatures is site/season specific, therefore, warming will be more severe in certain regions/seasons than in others [1,3]. Nepalese mountains are among the regions projected to experience a higher rate of warming than the global average [8]. Accordingly, the effects of climate change in the Nepalese Himalaya (Central Himalaya) are expected to be more pronounced than in other regions of the world [9,10]. Nevertheless, there are relatively fewer studies on the impacts of climate change in the Himalaya compared to other regions. Furthermore, most previous studies on climate change's impact on the Himalaya focus on the extent of glacial melting, alternation in precipitation patterns, and range shifts [10–14] while studies on the impact of climate change on biodiversity and ecosystem functioning are meagre.

Nepal's unique geography and climate harbours hundreds of valuable medicinal plants (MPs) with high market values both inside and outside of the country. Since prehistoric times, MPs have been the source of traditional health care system and income generation, thus playing a key role in the livelihood of many communities across the country [15–17]. In recent days, MPs not only contribute to the primary healthcare and livelihood of local communities but also contribute to the national economy through revenue and employment generation [18–20]. Therefore, Nepalese MPs have been integral to the healthcare system, livelihood, and national economy [16,17,19,20]. However, such high-valued MPs are under threat due to multifarious factors such as habitat loss associated with recent land use and land cover changes, immature harvesting, overharvesting pressure, and most importantly the recent climate change [21–24]. Although there is a plethora of studies on Nepalese MPs, most of these studies are primarily focused towards the various aspects of ethnobotany (documentation, use pattern phytochemistry, efficacy etc.) and economic valuation [16,18,20,24,25], ecological studies [26,27], a few are focused towards pollination biology [28–32] while comprehensive knowledge of how climate change would impact the distribution of a particular medicinal plant species is yet meagre. This suggests the need for multidimensional comprehensive studies for the long-term conservation and sustainable utilization of MPs.

*Rheum australe* D. Don (Polygonaceae), the Himalayan rhubarb, commonly known as Padamchal is one of the highly prioritized medicinal plants of Nepal. In many traditional medicinal practices such as Aayurveda, Homeopathic, Tibetan, and Unani, it is used extensively for the treatment of a wide range of ailments related to the digestive, circulatory, respiratory, endocrine, and skeletal systems, and for various infectious diseases [33,34]. In the modern medical system also, its demand is very high as the plant contains a variety of phytoconstituents, particularly anthraquinones derivatives and stilbene derivatives with well-known pharmacological efficacy against a spectrum of health ailments such as anti-inflammatory, anticancer, antidiabetic, antifungal, antimicrobial, antioxidant, and enhances Liver and Kidney health [33–35]. Therefore, the plant is vulnerable due to overharvesting pressure for trade. Both roots and petiole are in trade. Around 400 tons, including both roots and petiole, of *R. australe* contributing approximately USD 740,000 (0.0021% of national GDP, 12.15% of national revenue of the medicinal plant trade) are annually traded from Nepal [36]. Due to this high trade potential, its natural population in the wild is decreasing rapidly. Therefore, the loss of this important plant resource from the wild not only has ecological implications but also affects the local livelihood and national economy. However, we entirely lack data on its potential distribution range under current and future climate scenarios. This indicates a need for a comprehensive assessment of the impact of climate change on the distribution of *R. australe* under all the potential emission scenarios.

Species distribution modelling (SDM) is one of the widely used approaches to study the impact of climate change on the distribution of a taxon in a particular region/on a global scale [37,38]. SDMs are a class of methods that use occurrence data in conjunction with environmental data to make a correlative model of the environmental conditions that meet a species' ecological requirements and predict the relative suitability of habitat [37,39]. SDMs are widely used throughout the world because they are simple to use and highly cost-effective. Various SDMs such as Profile Methods (Bioclim, Mahalanobis distance), Classical Regression Models (GLM: generalized linear model, GAM: generalized additive model), Machine Learning Approaches (MaxEnt: maximum entropy, SVM: support vector machine, RF: random forest, ANN: Artificial neural network, BRT: boosted regression trees), and Ensemble models have been extensively used to estimate species potential distribution range for the current and future climate scenarios [23,40–42]. These SDMs are based on various assumptions and use their specific algorithms to estimate the distribution range of a taxon. Among the various models, Maximum Entropy (MaxEnt) is the most widely used approach for the SDMs. MaxEnt uses the principle of maximum entropy on presence-only data and computes a set of functions that relate environmental variables and habitat suitability to estimate the species' potential geographic distribution [43]. Moreover, MaxEnt produces robust predictions even when the sample size is small [44]. Therefore, MaxEnt has become one of the most popular tools for modelling species distribution, with hundreds of peer-reviewed articles published each year [45].

In this study, we used the MaxEnt modelling approach to predict the distribution range of *R. australe* in the geographically complex mountainous terrain of Nepal for the current and four future periods: 2021–2040, 2041–2060, 2061–2080, and 2081–210 under four potential emission scenarios: SSP126, SSP245, SSP370, and SSP585. In addition to distribution modelling, we also aimed to identify potential planting/long-term genetic resource conservation areas for *R. australe* in Nepal. The findings of this study would thus not only provide comprehensive information on the potential impact of climate change on this valuable MP but also have policy implications for the conservation and sustainable utilization of *R. australe* in Nepal.

## Materials and methods

### Study species

*Rheum australe* D. Don (Polygonaceae) is a Himalayan endemic perennial herb (Fig 1A) distributed in the sub-alpine regions between 2740–5200 m elevations [33,34,46]. The taxonomic description of this species is available elsewhere [33,34,46]. It grows in grassy or rocky slopes, crevices, between boulders, forest margins, and near streams. It is known to occur in the six Pan-Himalayan countries: Nepal, China, Bhutan, Myanmar, India, and Pakistan. In Nepal, it is distributed throughout the country between the elevation of 2740–4400 m [33,34].

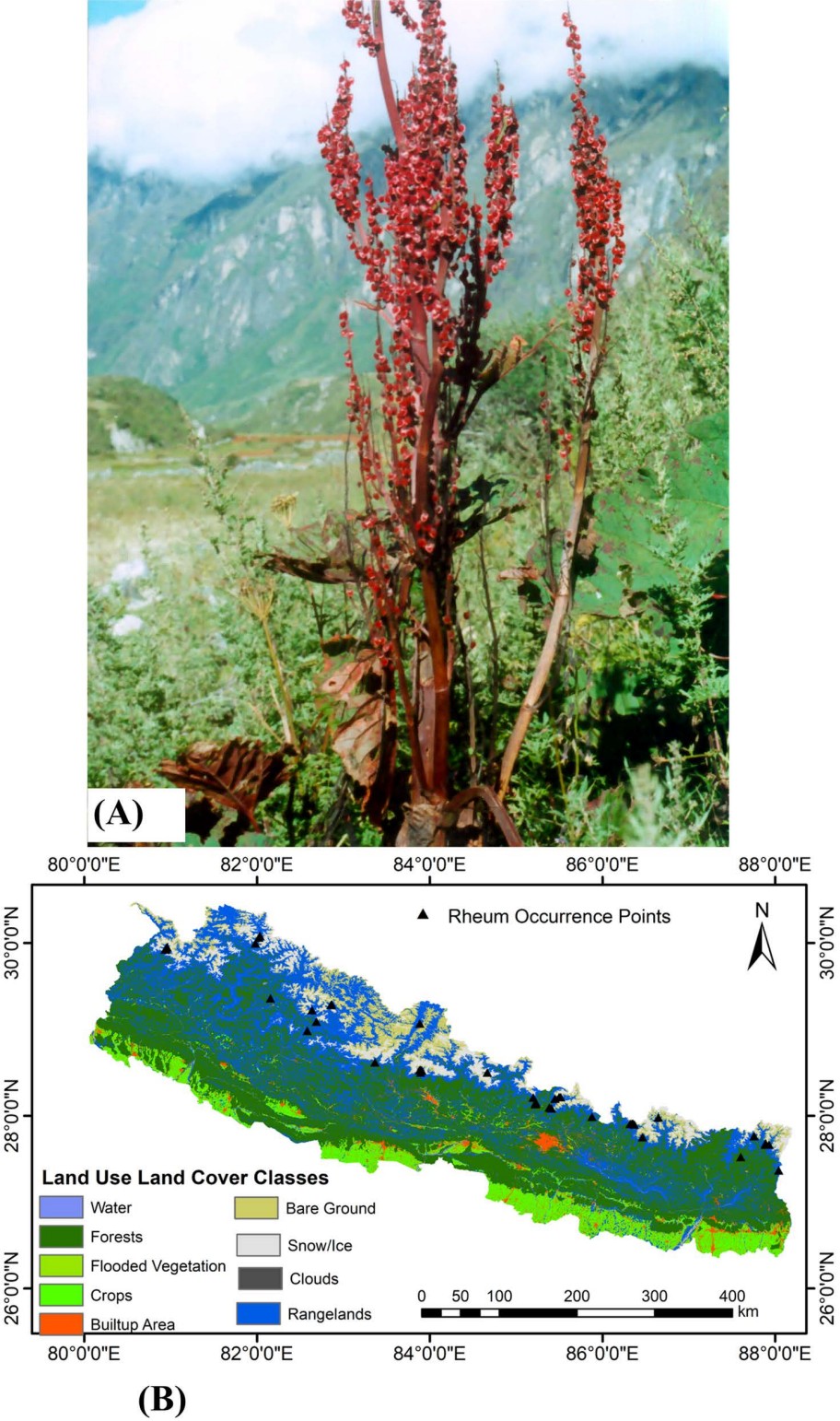

**Fig 1. Study species and its distribution range in Nepal.** A-*Rheum australe* at its natural habitat, B- Occurrence points of *R. australe* across different Land Use and Land Cover classes in Nepal.

## Natural distribution data

The distribution data of *R. australe* in Nepal were sourced from field surveys, a review of literature, herbarium studies from National Herbarium and Plant Laboratories (KATH) and Tribhuvan University Central Herbarium (TUCH), and through online databases including iNaturalists (www.inaturalists.org), the Global Biodiversity Information Facility (www.gbif.org) [47], floral of Nepal (www.floraofnepal.org), the herbarium at the University of Tokyo, Japan (https://umdb.um.u-tokyo.ac.jp), and the herbarium at the Royal Botanical Garden, Edinburgh, the United Kingdom (https://data.rbge.org.uk). The occurrence records of the species were accessed from the online databases on 20 June 2024. Through these sources, a total of 134 occurrence points were compiled which were mapped on a base map of Nepal to test the accuracy of the occurrence points. Incorrect and duplicate points were removed. Data rarefaction reduces the sampling bias and improves the predictive performance of the models [48], therefore, to minimize the spatial autocorrelation, the remaining points were rarefied to a $1km^2$ grid using the Humboldt package [49] in the R (version 4.2.2) [50]. After rarefaction, a total of 41 points were retained (Fig.1B, supplementary table: Table S1).

## Environmental variables

We considered 24 environmental variables comprising the combination of 19 bioclimatic variables and 5 topographic factors (table 1) as the predictors of habitat suitability of our study species because several previous studies have suggested the important role of these environmental variables for the distribution of plant species in mountain regions such as Nepal [23,51,52]. We downloaded 19 bioclimatic variables and elevation with a spatial resolution of 30s (*ca* 1 $km^2$) from the WorldClim database (https://www.worldclim.org/data/worldclim21) for the current periods [53] and future scenarios (https://www.worldclim.org/data/cmip6/cmip6_clim30s). For future climate scenarios, the Intergovernmental Panel on Climate Change (IPCC) has proposed four Shared Socio-economic Pathways (SSPs): SSP126, SSP245, SSP370, and SSP585 based on different socio-economic assumptions leading to varying carbon emissions levels [54,55]. The SSP126 scenario assumes that the radiative forcing decreases to 2.6W/$m^2$ and there will be a low green gas concentration. The SSP245 scenario assumes that the radiative forcing stabilizes to 4.5 W/$m^2$ and greenhouse gas concentration gradually increases but remains at relatively low levels. It has been estimated that in the emission scenario SSP370 the radiative forcing stabilizes to 7 W/$m^2$ but there will be relatively high carbon dioxide and other greenhouse gas concentrations. In the emission scenario SSP585, it has been speculated that the radiative forcing increases to 8.5 W/$m^2$ with the very high concentration of greenhouse gas representing a composite scenario of an energy-intensive socio-economic development path [54,55]. The updated Coupled Model Intercomparison Project CMIP6 data is available for the four Shared Socio-economic pathways (SSPs): SSP126, SSP245, SSP370, and SSP585 for the four different periods: 2021–2040, 2041–2060, 2061–2080, and 2081–2100. Likewise, several Global Circulation Climate Models (GCMs) have been proposed by various organizations [56]. Among them, we chose the data from Models for Interdisciplinary Research on Climate version 6 (MIROC6) because this model has consistent rainfall projections for the Indian sub-continent and thus performs better projections for the South-Asian regions than other available models [57,58]. Because the future scenarios are based on varying socio-economic assumptions and carbon emissions levels, in the context of Nepalese mountainous terrain, future scenarios are very uncertain. Thus, we selected all four SSPs for the four time periods to make the future predictions more comprehensive. In addition to bioclimatic variables and elevations, we also included land aspect, slope, evapotranspiration, and land use and land cover layers as predictors. The land aspect and slope were derived from the elevation data in ArcGIS 10.8 (ESRI, 2021). Evapotranspiration data was downloaded from the Consortium for Spatial Information, Consultative Group for International Agriculture Research (CGIAR-CSI) (https://cgiarcsi.community) while data on land use and land cover class was downloaded from the "awesome-gee-community-catalogue" S2TSLULC Project (https://gee-community-catalog.org/projects/S2TSLULC/) [59]. Thus, a total of 24 variables were considered as the predictors (Table 1) of habitat suitability for *R. australe* in Nepal. All the downloaded raster data were projected for the WGS1984 system, downscaled for Nepal using the extraction by mask function in ArcGis10.8 (ESRI, 2021), and converted into ASCII

**Table 1. Bioclimatic variables considered for the MaxEnt-based habitat modelling of *Rheum australe* in Nepal. Variables in boldface indicate the bioclimatic variables retained after correlation and Variance Inflation Factor (VIF) analysis which were used as the predictors for the modelling of habitat suitability of *R. australe* under current and future different climate scenarios.**

| Code | Environmental variables | VIF | Units |
|---|---|---|---|
| Bio_1 | Annual Mean Temperature | | °C |
| **Bio_2** | **Mean Diurnal Range (Mean of monthly (max temp – min temp)** | 3.280819 | °C |
| Bio_3 | Isothermally (BIO2/BIO7) (* 100) | | % |
| **Bio_4** | **Temperature Seasonality (standard deviation *100)** | 2.548957 | % |
| Bio_5 | Maximum Temperature of Warmest Month | | °C |
| Bio_6 | Minimum Temperature of Coldest Month | | °C |
| Bio_7 | Temperature Annual Range (Bio5-Bio6) | | °C |
| Bio_8 | Mean Temperature of Wettest Quarter | | °C |
| **Bio_9** | **Mean Temperature of Driest Quarter** | 2.225010 | °C |
| Bio_10 | Mean Temperature of Warmest Quarter | | °C |
| Bio_11 | Mean Temperature of Coldest Quarter | | °C |
| **Bio_12** | **Annual Precipitation** | 4.173975 | mm |
| Bio_13 | Precipitation of Wettest Month | | mm |
| **Bio_14** | **Precipitation of Driest Month** | 4.655152 | mm |
| **Bio_15** | **Precipitation Seasonality (coefficient of variation)** | 4.181939 | % |
| Bio_16 | Precipitation of Wettest Quarter | | mm |
| **Bio_17** | **Precipitation of Driest Quarter** | 5.269563 | mm |
| Bio_18 | Precipitation of Warmest Quarter | | mm |
| Bio_19 | Precipitation of Coldest Quarter | | mm |
| **Asp** | **Land aspect** | 1.414686 | ° |
| **Elv** | **Elevation** | 4.597771 | m |
| PET | Evapotranspiration | | mm/day |
| **LULC** | **Land Use Land Cover** | 1.434518 | Categorical |
| **SI** | **Slope** | 1.347936 | ° |

file format for projection in MaxEnt. High correlation and collinearity among the predictors could cause overprediction and thus affect the accuracy of the predictions. Therefore, we performed correlation and Variance Inflation Factor (VIF) analysis to test the co-linearity among the 24 environmental variables. Variables with VIF ≥ 10 were considered as highly correlated. 13 variables: bio11, bio18, bio1, bio3, bio5, bio6, bio7, bio8, bio10, bio13, bio16, bio19, and evapotranspiration from the 24 input variables have collinearity problems. After excluding the collinear variables, only 11 variables were retained with the linear correlation coefficients ranging between -0.00008 (elevation~bio17) to 0.7993978 (bio17~bio14). The retained 11 variables (Table 1) were used as the predictors for the modelling of habitat suitability of *R. australe* under current and future different climate scenarios.

## Construction of Maximum Entropy (MaxEnt) model

We used MaxEnt java (https://biodiversityinformatics.amnh.org/open_source/maxent/) version 3.4.4 [60] to construct the distribution models of *R. australe* under current and various future climate scenarios. The rarified and cleaned geographic distribution data of *R. australe* was converted into a CSV file format and uploaded as samples into the Maxent software. The 11 variables (Table 1) were used as the environmental layers while the same 11 variables for the four different SSPs and four different periods: 2021–2040, 2040–2060, 2060–2080, and 2081–2100 were included as the projection layers. Therefore, a total of 17 projection layers were considered for the model construction. The bias file was prepared using the

selection by polygon/map algebra function in ArcGIS 10.8 by including only the local units (rural municipalities/municipalities) with the presence data for *R. australe* as the polygons. The maximum number of background points was set to 10,000. To avoid the potential shortcomings of the MaxEnt model, we executed the model by optimizing various parameters following Warren (2011) and Merow (2013) [61,62]. To deal with model complexity, we selected auto-feature, create response curves, and make pictures of predictions in the MaxEnt GUI (graphic user interface). We used jackknife analysis to measure the relative contribution of various environmental factors in the model development. The output format was selected as Logistic and the output file type as ASCII. We set the maximum number of iterations to 5,000; the convergence threshold to 0.00001; and the regularization multiplier to 0.1 while the rest of the parameters were set to default. This setup enhances the predictive accuracy and stability of the MaxEnt, and thus the predicted models are robust [61–64]. We then ran the model by considering 25% of the data as the test data (for model validation) while 75% data was used as the training data (for model calibration). We selected the subsample as the replicate run type and executed 30 replicate runs. Finally, the results of all 30 models were averaged. The models' robustness was enhanced by using 10-percentile training presence threshold criteria to determine the cutoff values of the habitat suitability. The averaged outputs of the 30 models for each scenario were processed in ArcGIS to generate the outputs in raster format for further analysis.

## Evaluation of the model

We tested the accuracy of the models by using two different measures: True Skill Statistics (TSS) and area under the curve (AUC) values of the receiver operator characteristic (ROC). These two values are based on different measures and characterize different weights to the various types of prediction errors. TSS is a threshold-dependent measure of model accuracy. Its values range from −1 to +1 where positive values indicate agreement between prediction and observation while zero or negative values indicate that the agreement is more likely to be by random classification [65]. In general, values less than 0.4 indicate poor performance of the model, 0.4 to 0.8 are useful while values greater than 0.8 indicate the excellent performance of the model [66,67]. AUC is independent of the threshold and the frequency of occurrence of the target species with values ranging from 0 to 1 [68]. Following the classification of Rana et al. (2017) and Franklin et al. (2010) [66,69], we considered the model performance poor if the AUC value was less than 0.7, while the performance of the models with AUC values between 0.7 to 0.9 was considered moderate and the models with AUC values greater than 0.9 were considered excellent.

## The relative contribution of environmental variables in the model prediction

To determine the relative contribution of environmental variables in the model prediction, we considered the jackknife test of regularized training gain and also included an analysis of variable contributions. Moreover, we also considered the response curves to determine the optimum condition of each environmental variable for the distribution of *R. australe* in Nepal.

## Classification of the potential habitat of *Rheum australe*

We imported the average outputs of the MaxEnt models for each projection scenario into ArcGIS for the classification of the potential habitat of *R. australe*. Maxent produces a continuous raster with threshold values from 0 (least) to 1 (highest) suitability zones. Further classification of the model output into different habitat suitability classes could be performed by determining a threshold value. Following Phillips (2006) [43], we chose the 10-percentile training presence logistic threshold to determine the minimum threshold value for delineating the suitable habitat for *R. australe* into different suitability zones. Here, the grids with threshold values less than the 10-percentile training presence logistic threshold were considered unsuitable while the rest areas were considered suitable habitats of *R. australe*. The suitable habitat zones were further classified using Jenk natural classification into three different categories: low suitable (Threshold value ≤ 0.4, i.e., up to 40% probability of occurrence), medium suitable (Threshold value between 0.4 to 0.6), and high suitable (Threshold

value > 0.6) following the criteria used by Rawat et al. (2022) and Yang et al. (2013) [67,70]. We then estimated the area of each suitable zone in the raster package in R 4.2.2 [50]. To identify suitable districts for the current distribution and also to assess the potential impact of various future climate scenarios on the district-wise distribution of *R. australe,* the predicted habitat models were projected into the district levels.

### Change of the potential habitat of *Rheum australe* under different climate scenarios

To determine the potential changes in the suitable habitat of *R. australe* under the currently considered future climate scenarios and periods, we calculated the area of the suitable habitat of each scenario/period. For this calculation, the area with threshold values greater than 10-percentile training presence logistic threshold was considered suitable. Then, we calculated the difference in the suitable habitat area between the future climate scenarios and the current suitable habitat. We also assessed the area of suitable habitat for *R. australe* in each suitable district both under the current and future climate scenarios and calculated the difference in the suitable habitat area between the future climate scenarios and the current suitable habitat. The area calculation was performed in the raster package in R 4.2.2 [50].

### Identification of potential planting area

The current distribution records, our observation in the field throughout the country over the last several years, and recent studies on the habitat assessment reveal that forests and/or rangelands are the current habitats of *R. australe,* i.e., it is not found occurring in the land types other than forests/rangelands. Therefore, we considered forests and rangelands as the ideal habitat for planting and conservation of *R. australe*. To determine if the projected suitable habitats of *R. australe* overlap with the forests and rangelands, we imported the maxent prediction of suitable habitats of *R. australe* into ArcGis and performed an overlay analysis with the current (2023) land use and land cover (LULC) data consisting nine land use classification categories (Water, Flooded Vegetation, Forest, Cropland, Builtup Area, Bare Ground, Rangelands, Snow/Ice, and Cloud- No landcover information due to persistent cloud) downloaded from the "awesome-gee-community-catalogue" S2TSLULC Project (https://gee-community-catalog.org/projects/S2TSLULC/) [59]. However, it revealed that the maxent predicted suitable habitats of *R. australe,* both for the current condition and future scenarios, also occurred beyond forests and rangelands. Therefore, as we considered only the forests and rangeland as potential areas for planting and long-term genetic resources conservation, the MaxEnt predicted suitable habitats lying beyond forests and rangelands were considered unsuitable and excluded for further analysis. We again performed spatial overlay analysis between forests/rangelands and the maxent predicted habitat of *R. australe* under current and future different climate scenarios. The areas/regions belonging to the highly suitable habitat class and lying in forests/rangelands have been considered the suitable habitat of *R. australe* and potential planting/ long-term genetic resources conservation areas. We then estimated the area of suitable habitat under current and future climate scenarios. To further identify the most appropriate districts for plantation/long-term genetic resources conservation, we estimated the area of suitable habitat at each suitable district level and also assessed the potential changes in the suitable area under different climate scenarios. The districts having relatively higher suitable areas for the current condition that are expected to have increased suitable areas for the future scenarios are considered as the best potential planting/conservation sites. On the other hand, the districts with the potentiality of losing a majority of suitable habitat in future climate scenarios even if they have higher suitable areas at the present condition are considered unsuitable for planting/developing conservation sites.

## Results

### Model evaluation and relative contribution of environmental variables

The receiver operating characteristic (ROC) curve of the maxent model for training data was 0.921 which indicates the excellent performance of the model when AUC alone is considered as an index of the model performance [66,69]. The True Skill Statistics (TSS) was 0.68 which indicates the moderate (useful), not excellent, performance of the Model.

However, several previous studies have considered the model with the TSS value > 0.6 valid for predicting the distribution of plants in mountainous regions [23,66,67,69]. Therefore, given the high value of AUC and TSS > 0.6, we considered the model valid for predicting the distribution of *R. australe* under the current and future climate scenarios. The mean value of the AUC of the ROC obtained after 30 replicate runs was 0.807 with a standard deviation of 0.044 (Fig 2A). This AUC value is greater than 0.5 of a random prediction and thus indicates the accurate and precise performance of the Maxent model in predicting the suitable habitats of *R. australe* in Nepal.

The result of the Jackknife test (Fig 2B) revealed that elevation contributed the highest gain when used in isolation followed by bio12 (annual precipitation), bio17 (precipitation of driest quarter), and bio14 (precipitation of driest month). Therefore, elevation, bio12, bio17, and bio14 are the most important environmental variables influencing the current geographic distribution of *R. australe* in Nepal. The slope is the environmental variable that decreases the gain the most when it is omitted, which therefore appears to have the most information that is not present in the other variables. The analysis of variable contribution revealed that bio12, bio14, slope, land use/land cover, land aspect, and elevation were the major contributing factors to the model development (Table 2). The environmental variable "bio12" contributed the highest (32.7%) followed by bio14 (17.8%), slope (10.2%), land use land cover (9%), aspect (8.3%), and elevation (8.3%). These six environmental variables contributed a total of 86.1% in the model development and thus suggest that these six environmental factors are the major limiting factors for the potential distribution of *R. australe* in Nepal Himalaya. In terms of permutation importance, bio12 (33.7%), bio14 (17.7%), elevation (11.4%), and land aspect (9.4%) were found as the four most important environmental variables indicating greater reliance of the model upon these four variables (Table 2).

The result of the response curves revealed that the probability of distribution of *R. australe* is highest when the precipitation of the driest month is around 10 mm (Fig. 3C). The probability of distribution is highest when the annual precipitation is around 600 mm and decreases with the increase in annual precipitation. Likewise, the precipitation seasonality of around 35 mm is the optimum condition for the distribution of *R. australe*. The probability of distribution increases with increasing precipitation of the driest quarter up to 110 mm and then tapers off. The probability of distribution is highest at around 25˚ land aspect and decreases sharply after that. The mean diurnal air temperature range between 11˚C and 11.5˚C is the optimum condition for the distribution of *R. australe*, after this point, the probability decreases sharply. The probability of distribution increases gradually with the rise in temperature seasonality which peaks off at 75% (Fig. 3G). The probability of distribution is expected to be high when the temperature of the driest quarter is at around - 5˚C. We found that the elevation range between 3000-4000m represents the suitable habitat zone with an elevation of *ca* 3500 m as the optimal zone for the distribution of *R. australe* (Fig. 3I). Likewise, the probability of distribution is highest when the slope of the habitat is around 5 ˚ to 8 ˚; above this slope, the likelihood of distribution decreases sharply (Fig. 3J). Overall, the result reveals that the distribution of *R. australe* is optimum in a north/northeast facing landscape at an elevation of around 3500m that has very low steepness (slope below 10 ˚) and receives around 600 mm annual precipitation with around 10 mm precipitation during the driest month.

## The geographic distribution of *Rheum australe* under current and future climate scenarios

The geographic distribution of *R. australe* under the current (1970–2000) climate scenario revealed a total of 22,318.66 km², 15.13% of the total area of Nepal, suitable for the distribution of *R. australe*. Of the total suitable areas, 14,500.37 km² (9.83%) is found least suitable, 6,798.44 km² (4.61%) medium suitable, and 1,019.86 km² (0.69%) the most suitable habitat for *R. australe.* The suitable area for all future scenarios is expected to decrease with the highest decrease for SSP585:2081–2100 (34.59%); while for scenario SSP126:2021–2040, the decrease in suitable area is expected to be the least (6.08%) (Table 3). Both for the current and future climate scenarios, the area of suitable habitat is greater in the western region of the country than in the central and eastern regions (Figs 4–6).

On the district level, among the 77 districts, only 25 districts from the mid-hills/higher mountains of Nepal have suitable habitats for the distribution of *R. australe* (Fig 4). For the current climatic conditions, the area of probable occurrence is

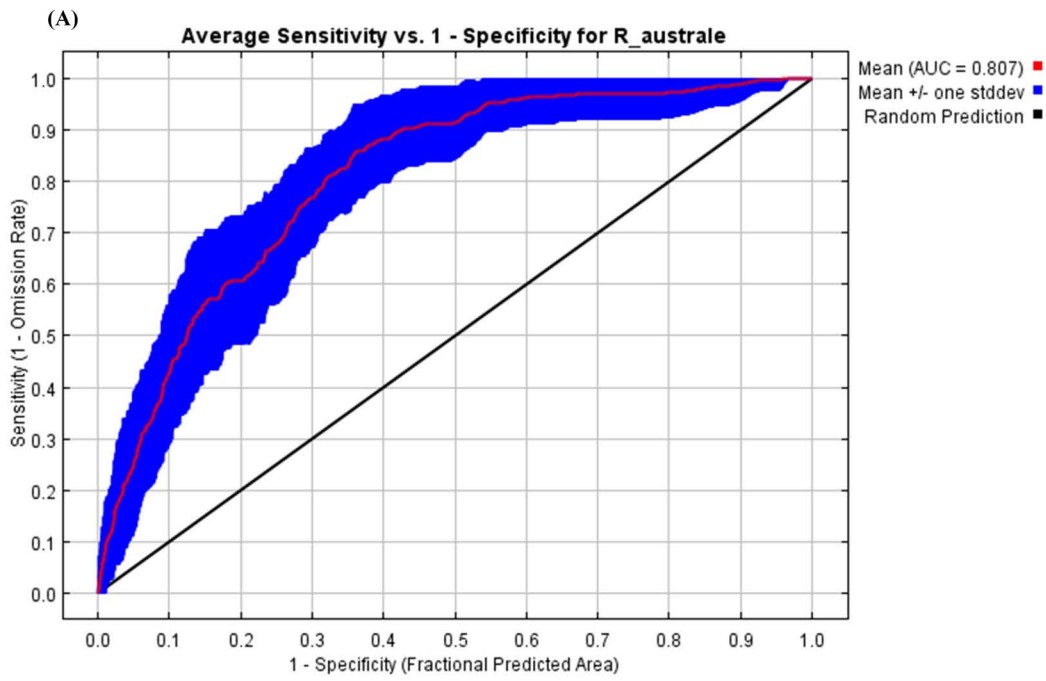

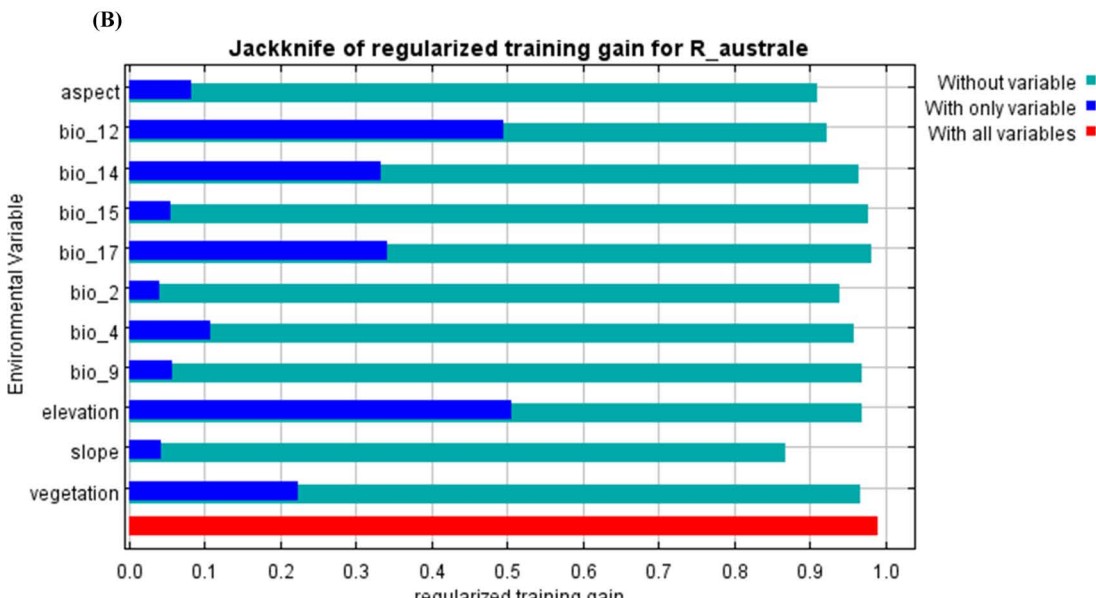

**Fig 2. Model performance and the key environmental variables affecting the distribution range of *Rheum australe* in Nepal.** A- ROC curve of the MaxEnt result, B- The jackknife test result for the environmental variables.

highest in the Humla District followed by Darchula, Bajhang, Jumla, and Baglung Districts. In contrast, Rukum-West District has the least suitable area followed by Jajarkot, Mustang, Lamjung, and Ramechhap Districts (Table 4). Likewise, for the different future climatic sceanrios, the distribution pattern is consistent with the current climatic conditions; the same 25 districts only possess suitable habitats for the distribution of *R. australe* (Table 4). However, five districts are expected

**Table 2. Estimates of variables' relative contribution and permutation importance in developing the distribution model of *Rheum australe* in Nepal.** The environmental variables in the table are arranged in the order (high to low) of their contribution (%) to the model development.

| Environmental variables | Contribution (%) | Permutation importance (%) |
|---|---|---|
| Annual precipitation | 32.8 | 33.7 |
| Precipitation of Driest Month | 17.8 | 17.7 |
| Slope | 10.2 | 4.5 |
| Land Use and Land Cover | 9 | 1 |
| Aspect | 8.3 | 9.4 |
| Elevation | 8.3 | 11.4 |
| Precipitation of Driest Quarter | 4.6 | 0.7 |
| Mean Diurnal Range | 3.9 | 6 |
| Mean Temperature of Driest Quarter | 2.5 | 5.1 |
| Temperature Seasonality | 1.5 | 7.2 |
| Precipitation Seasonality | 1.1 | 3.3 |

to have increased suitable areas, 10 districts are expected to have decreased suitable habitats, while the rest districts are expected to have inconsistent change in suitable habitats (increased for some scenarios/periods and decreased for some scenarios/periods) under different future climatic conditions (Table 4). Remarkably, all the districts are expected to lose the highest suitable area in scenario: SSP585:2081–2100 while for the climatic scenario: SSP126: 2021–2040, the decrease in the suitable area is the least (Table 4).

### Identification of potential planting area

The result revealed that the overall (at the country level) suitable habitat (the high suitable habitat lying only within forests and rangelands) of *R. australe* under the current climatic condition is 937.99 km$^2$ indicating that 81.87 km$^2$ projected high suitable habitat lies in other land-use types (Table 5). Compared to current climatic conditions, the suitable area of *R. australe* is expected to decrease under all the future climate scenarios/ periods except for scenario SSP245 from 2081 to 2100 in which the suitable area of *R. australe* is expected to increase by 3.67% (Table 5). The highest decrease (54.92%) in suitable habitat is anticipated for scenario SSP585 from 2081 to 2100 while for scenario SSP126 from 2021 to 2040, the decrease in suitable habitat is expected to be the least (5.12%) (Table 5).

The district-wise suitable habitat of *R. australe* both for the current and future climatic scenarios is presented in Table 4 while the district-wise changes in the suitable habitat under different future climatic scenarios are presented in Table 6. The result revealed that in the Jumla district, the suitable habitat of *R. australe* is expected to increase in all the currently considered future climate scenarios (Table 6). Likewise, districts: Kalikot, Dolpa, Gorkha, and Jajarkot are expected to have increased suitable habitat for almost all (except one, two or three scenarios/periods) future climate scenarios and periods (Table 6). In contrast, districts: Humla, Darchula, Bajhang, Myagdi, Dolakha, Rasuwa, Lamjung, and Kaski are expected to have decreased suitable habitats for all future climate scenarios and periods (Table 6). Likewise, districts: Sindhupalchok, Baglung, and Mustang are expected to have decreased suitable habitat for almost all (except one) future climate scenarios and periods (Table 6). Among the 25 districts, Humla will lose the highest suitable habitat of *R. australe* in future climatic scenarios followed by Darchula and Bajhang.

### Discussion

In recent years, several studies have used species distribution modelling approaches to predict the impact of climate change on the future distribution of Nepalese MPs under the projected climate change scenarios [23,51,66,71–74]. Given that the accuracy of a model prediction is greatly influenced by several factors such as the environmental variables

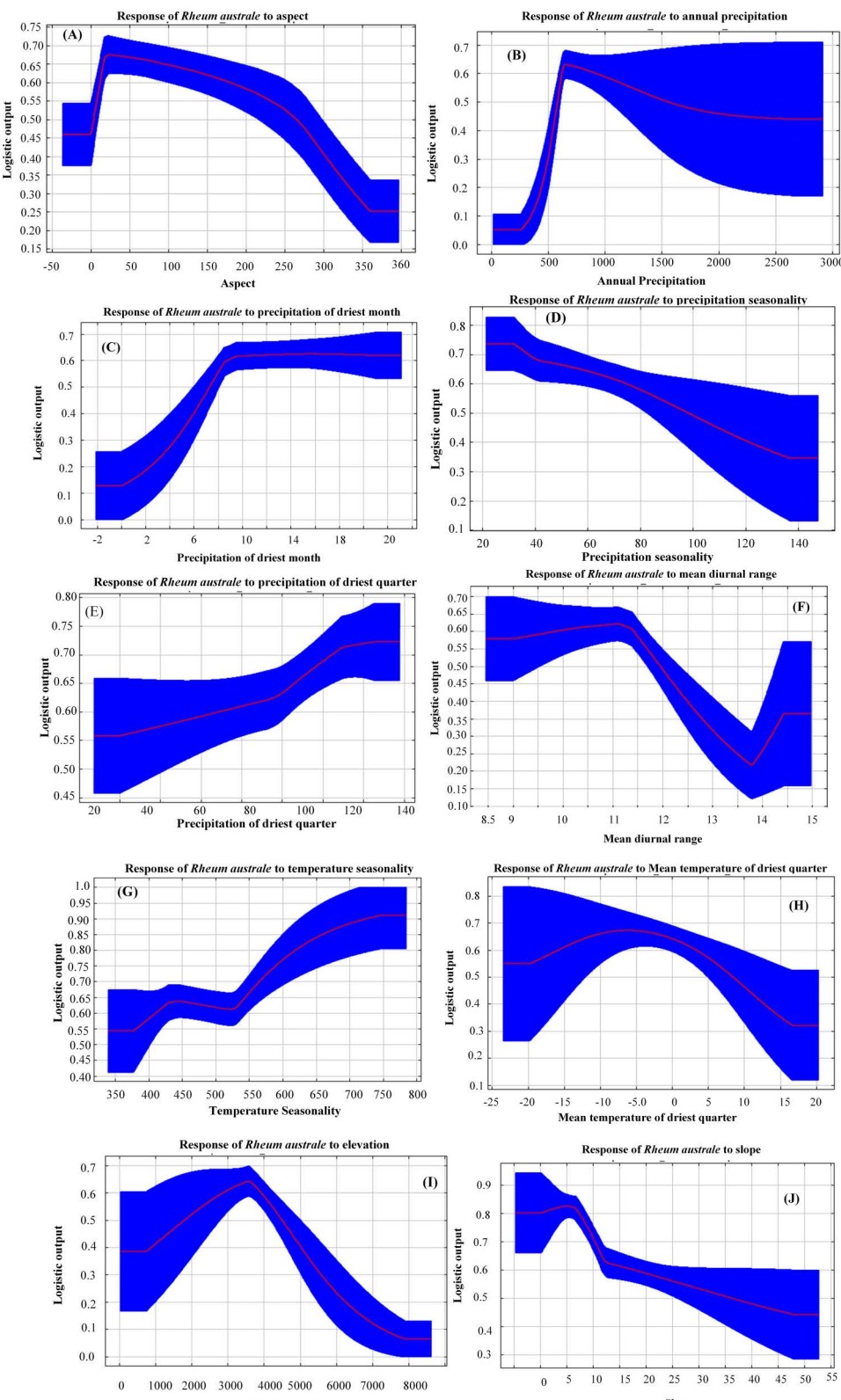

**Fig 3. Response curves of the key environmental variables showing the Logistic output for *Rheum australe*.**

**Table 3. Area of suitable habitat (including all land use types) of *Rheum australe* under current and future climate scenarios in the entire country context.**

| Socio-economic Pathways (ssp) and Time Periods | Total suitable area (km²) | Area Decrease (km²) | Low suitable area (km²) | Medium suitable area (km²) | High suitable area (km²) |
|---|---|---|---|---|---|
| Current (1970-2000) | 22318.66 (15.13%) | | 14500.37 (9.83%) | 6798.44 (4.61%) | 1019.86 (0.69%) |
| ssp126: 2021–2040 | 20961.05 (14.21%) | 1357.61 (6.08%) | 13965.68 (9.47%) | 6044.85 (4.10%) | 950.52 (0.64%) |
| ssp126: 2041–2060 | 19858.43 (13.46%) | 2460.23 (11.02%) | 13417.88 (9.1%) | 5589.99 (3.79%) | 850.56 (0.58%) |
| ssp126: 2061–2080 | 19400.74 (13.15%) | 2917.92 (13.07%) | 13206.63 (8.95%) | 5374.96 (3.64%) | 819.15 (0.56%) |
| ssp126: 2081–2100 | 19088.76 (12.94%) | 3229.91 (14.47%) | 13288.27 (9.01%) | 5027.05 (3.41%) | 773.44 (0.52%) |
| ssp245: 2021–2040 | 20559.36 (13.94) | 1759.30 (7.88%) | 13870.93 (9.40%) | 5809.71 (3.94%) | 878.71 (0.60) |
| ssp245: 2041–2060 | 19705.49 (13.36%) | 2613.17 (11.71%) | 13224.89 (8.97%) | 5567.14 (3.77%) | 913.46 (0.62%) |
| ssp245: 2061–2080 | 17940.14 (12.16) | 4378.52 (19.62%) | 12302.71 (8.34%) | 4818.55 (3.27%) | 818.88 (0.56%) |
| ssp245: 2081–2100 | 16784.19 (11.38%) | 5534.47 (24.80%) | 11039.74 (7.48%) | 4712.03 (3.19%) | 1032.42 (0.70) |
| ssp370: 2021–2040 | 20615.75 (13.98%) | 1702.92 (7.63%) | 13655.71 9.26%) | 5975.62 (4.05%) | 984.42 (0.67%) |
| ssp370: 2041–2060 | 19496.21 (13.22%) | 2822.45 (12.65%) | 13072.74 (8.86%) | 5486.26 (3.72%) | 937.22 (0.64%) |
| ssp370: 2061–2080 | 16376.88 (11.10%) | 5941.78 (26.62%) | 11548.20 (7.83%) | 4146.70 (2.81%) | 681.99 (0.46%) |
| ssp370: 2081–2100 | 16574.16 (11.24%) | 5744.51 (25.74%) | 11917.03 (8.08%) | 4035.20 (2.74%) | 621.92 (0.42%) |
| ssp585: 2021–2040 | 20205.49 (13.7%) | 2113.17 (9.47%) | 13446.11 (9.12%) | 5805.47 (3.94%) | 953.92 (0.65%) |
| ssp585: 2041–2060 | 18040.77 (12.23%) | 4277.90 (19.17%) | 12406.02 (8.41%) | 4813.59 (3.26%) | 821.15 (0.56%) |
| ssp585: 2061–2080 | 16098.25 (10.91%) | 6220.41 (27.87%) | 11519.49 (7.81%) | 3962.81 (2.69%) | 691.74 (0.42%) |
| ssp585: 2081–2100 | 14598.09 (9.90%) | 7720.57 (34.59%) | 10779.90 (7.31%) | 3348.06 (2.27%) | 470.14 (0.32%) |

and their spatial resolution, the accuracy of the species occurrence data, sampling biases, climate models, uncertainty of future climate scenarios, and the modelling approaches implemented, the findings should be interpreted cautiously [64,75]. Most of the previous findings are based on either the fifth version of the Coupled Model Intercomparison Project (CMIP5) data or even older CMIP data which are now considered obsolete and are frequently criticized for having several considerable biases [76]. Moreover, most previous studies assessed the impact of climate change based on either one or two emission scenarios for a few specific periods such as 2050 and/ or 2070. Indeed, predictions based on a few emission scenarios and relatively less reliable data, particularly in landscapes with extreme topographic heterogeneity as in the Nepalese mountainous terrain are inevitably ambiguous. Furthermore, studies have suggested that in addition to bioclimatic variables, the inclusion of additional abiotic/topographic variables such as elevation, aspect, slope, evapotranspiration, land use land cover data, and plant phenology would greatly enhance the performance of the models [42,77].

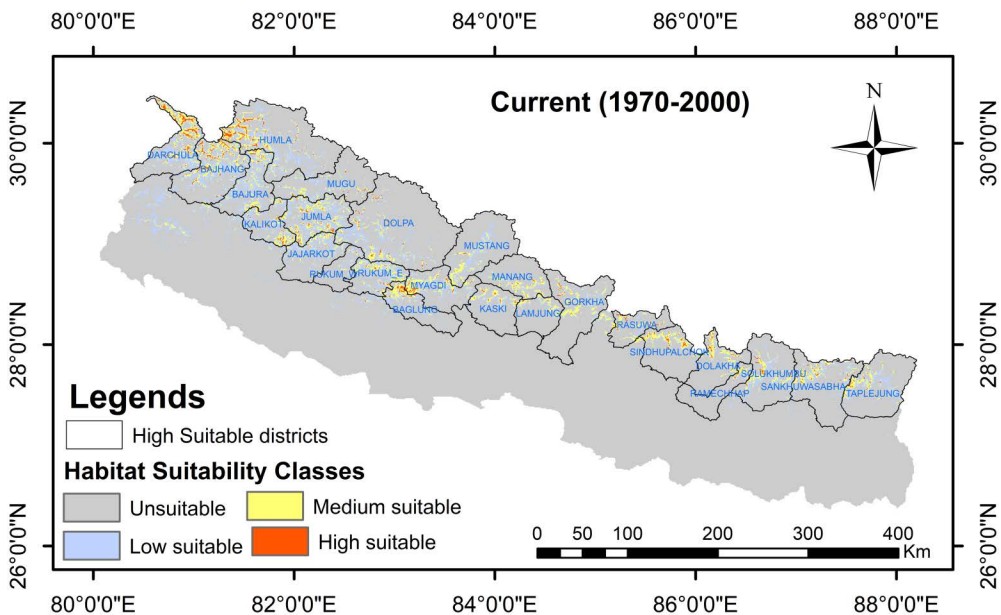

**Fig 4. The potential distribution range of *Rheum australe* in Nepal under the current climatic conditions along with the high suitable districts identified by the MaxEnt modelling.** Following Phillips (2006), MaxEnt predicted habitats with threshold values below the 10-percentile training presence logistic threshold were considered unsuitable (gray regions in the map). The remaining habitat zones were further classified into low suitable regions (threshold values ≤ 0.4; sky blue region in the map), medium suitable regions (threshold values between 0.4 to 0.6, yellow regions in the map), and high suitable regions (threshold values > 0.6, red regions in the map).

However, most previous studies on Nepalese MPs are almost entirely based on bioclimatic data only. Therefore, we still lack robust and comprehensive predictions on the impact of climate change on the distribution patterns of many valuable medicinal plants in Nepal with direct implications for local livelihoods and the national economy.

In this study, we assessed the distribution pattern of *R. australe* through the MaxEnt model because of its high predictive performance over other available SDM models. To avoid the potential shortcomings of the Maxent model such as overfitting, during model execution, we optimized all parameters following Warren et al (2011)[61] and Merow et al (2013) [62](Details presented in the method section). To reduce the shortcomings of the model performance associated with data accuracy, the occurrence points obtained through the online database were validated with the physical inspection of the herbaria from the National Herbarium and Plant Laboratories (KATH) and also with our multiyear field observation across the country. The sampling bias of the occurrence points was reduced by performing a spatial autocorrelation within 1 km, i.e., when there were two or more points within a 1 km distance, we considered only a single occurrence point for the model development. We incorporated 24 various predictors comprising the combination of classical bioclimatic variables from the current/CMIP6 database and additional topographic factors such as land use and land cover, potential evapotranspiration, elevation, land aspect, and slope. To capture the heterogeneity of the study site to a greater extent, we used the highest available spatial resolution data (*ca* 1km$^2$) [53] for model development. Studies suggest that compared to CMIP5 and older version CMIP data, CMIP6 data capture precipitation data reasonably well and have significantly low model bias. Particularly CMIP6 models are found robust in representing the spatiotemporal pattern of summer monsoon in China, Southeast Asia, Indian landmass and the North-East foothills of the Himalaya [58,76,78,79]. Therefore, CMIP6 data would produce robust future projections, at least in the context of the Nepalese Himalaya. Here, we selected MIROC6 data because it performs better projections for the South-Asian regions than other available climate models [57,58]. Moreover, considering the uncertainty of the future emission scenarios, we incorporated all four potential

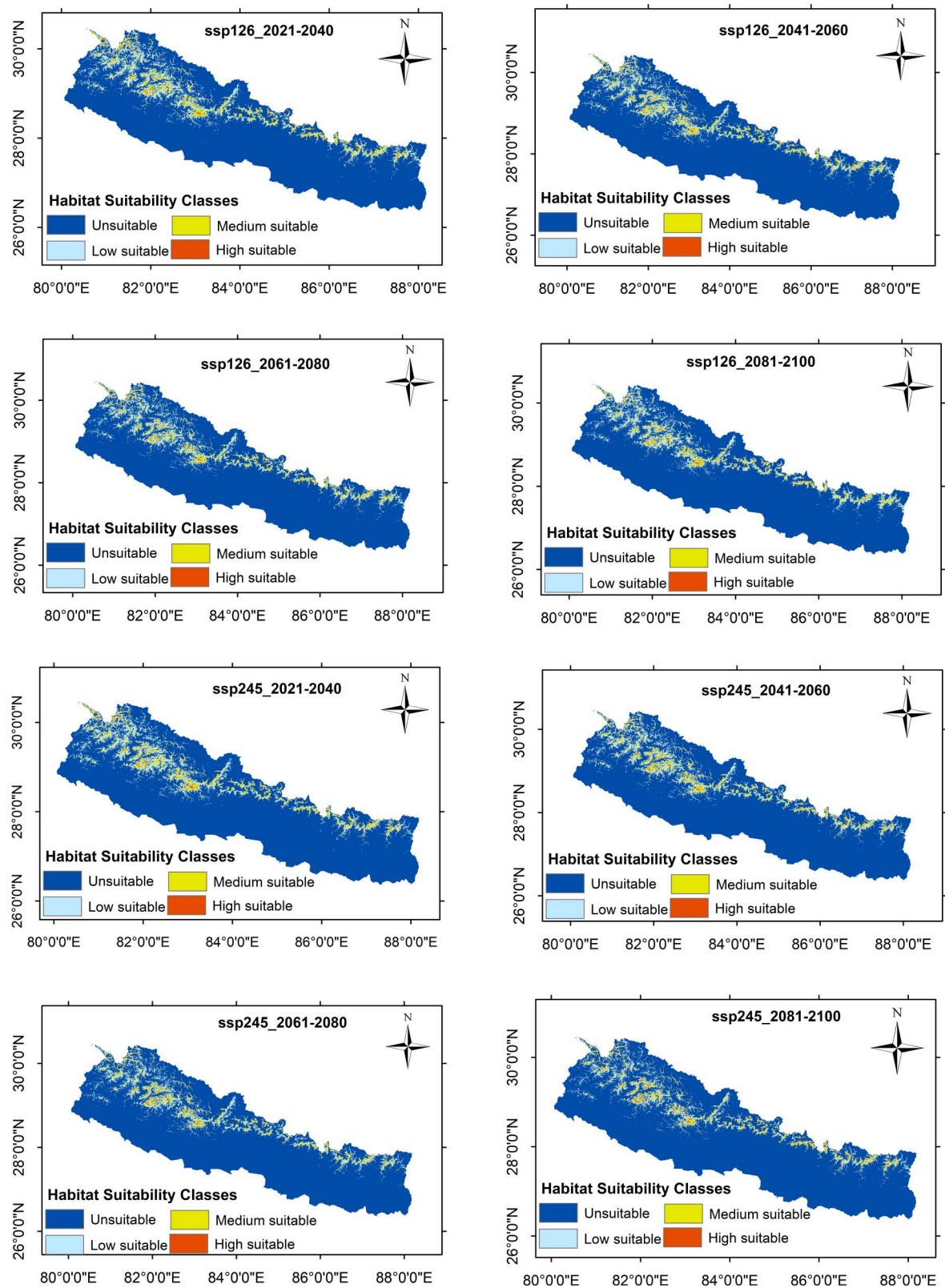

**Fig 5. The potential distribution range of *Rheum australe* in Nepal under the two future climate scenarios: SSP126 and SSP245 for the period 2021-2040, 2041-2060, 2061-2080, and 2081-2100.** Following Phillips (2006), MaxEnt predicted habitats with threshold values below the 10-percentile

training presence logistic threshold were considered unsuitable (dark blue regions in the map). The remaining habitat zones were further classified into low suitable regions (threshold values ≤ 0.4; sky blue region in the map), medium suitable regions (threshold values between 0.4 to 0.6, yellow regions in the map), and high suitable regions (threshold values > 0.6, red regions in the map).

socioeconomic pathways scenarios suggested by the IPCC for the four different periods from 2021 to 2100 AD [54,55]. Our result comprises 17 different potential projections and thus constitutes a comprehensive account of the potential impact of climate change on the distribution of one of the highly traded medicinal plants of Nepal for all the potential future emission scenarios for nearly a century (including the current distribution model and from 2021–2100 AD). Therefore, this in-depth approach distinguishes our study from existing literature on the habitat distribution modelling of Nepalese medicinal plants. Furthermore, considering our insightful attempt to minimize the shortfalls of the model prediction by optimizing each of the factors that affect the predictive performance of the model, we are quite confident that the predicted models we present here are robust within the premises of the modelling science. In the below paragraphs, we discuss the major implications of our current findings.

Our result reveals that elevation and precipitation-related factors such as annual precipitation, precipitation of the driest quarter, and precipitation of the driest month are the key factors in determining the distribution of *R. australe*. Our findings, the distribution pattern of *R. australe* influenced mainly by elevation and precipitation-related factors substantiate that this species is sensitive to elevation-related microclimate. Given that the driest month/quarter is the growing season of *R. australe,* and it preferably grows in areas with less water retention potentiality such as rocky slopes, crevices, and between boulders [34], this finding substantiates that sufficient precipitation in the driest month/quarter is a key requirement for its growth, development, and distribution. Like our current findings, precipitation-related factors play a key role in determining the distribution pattern of *R. webbianum* [80] and several medicinal plants in the mid-hills and high mountains of Nepal [23,51,66,81] and Indian western Himalaya [67,82,83]. In this regard, our result corroborates with several previous findings that elevation and annual precipitation play important roles in determining the distribution pattern of many plant species both globally and locally [42,63,66,81,84]. However, our result is inconsistent with several previous findings on Himalayan medicinal plants including two *Rheum* species (*R. alexandrae* and *R. nobile*) which suggest that temperature-related climatic factors contribute the most to their distribution [23,51,71,85]. Given the contrasting findings on the role of temperature and precipitation in determining the suitability zones of medicinal plants in the Nepalese Himalaya, it suggests that the response is individual species specific depending upon various intrinsic factors such as overall physiology of the plant, phenology and extrinsic factors such as biotic interactions that determine the species persistence and fitness through various mechanisms [38,86,87].

Studies have suggested that altitudinal or latitudinal range shift is the main detectable response of a plant species in escaping from the adverse effects of climate change [88–91]. Indeed, short distance assisted range expansion is the easiest strategy for a plant species to cope with the adverse effects of climate change [92,93]. Consistent with these previous findings, several herbaceous medicinal plants of the Himalaya including two *Rheum* species (*R. alexandrae* and R. *nobile*) are found migrating upward to mitigate the impact of climate change [85,89,94–96]. Therefore, we also speculated the upward migration of our study species in response to future climate change and increased suitability zones for *R. australe* in all future climate scenarios because most rangelands in the upper mountainous region could likely be transformed into suitable habitats. However, contrary to our expectations, the habitat suitability of *R. australe* is found to be decreased in all future climate scenarios. Although the suitable habitat of *R. australe* is anticipated to decrease under future climate scenarios, it is seen that the future distribution pattern is likely to be consistent with the current (historical) distribution pattern: The higher tendency of suitability zones in the western regions than in the central and eastern regions and the same 25 districts throughout the country represent the suitable habitat both for the current and future conditions (Figs 4-6). Given the decreased but consistent pattern of suitability zones among all the scenarios, it is less likely that *R. australe* move and

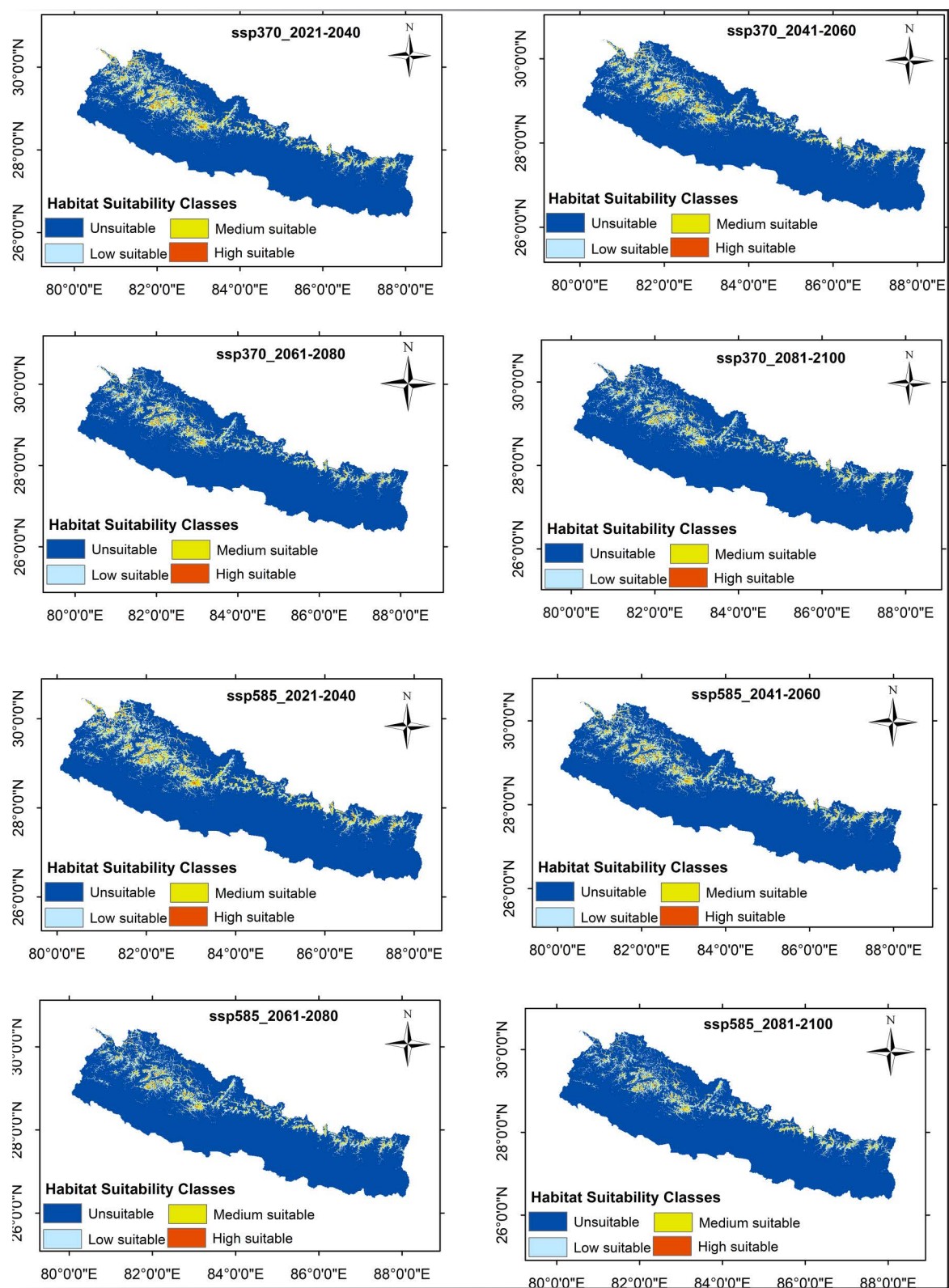

**Fig 6. The Potential distribution range of *Rheum australe* in Nepal under the two future climate scenarios: SSP370 and SSP585 for the periods 2021-2040, 2041-2060, 2061-2080, and 2081-2100.** Following Phillips (2006), MaxEnt predicted habitats with threshold values below the 10-percentile training presence logistic threshold were considered unsuitable (dark blue regions in the map). The remaining habitat zones were further

classified into low suitable regions (threshold values ≤ 0.4; sky blue region in the map), medium suitable regions (threshold values between 0.4 to 0.6, yellow regions in the map), and high suitable regions (threshold values > 0.6, red regions in the map).

inhabit the greater elevations in response to the predicted climate change, suggesting that land topography also has an important role in determining the distribution of *R. australe*. Similar to our finding, three *Rheum* species (*R. alexandrae*, *R. nobile* and *R. webianum* [80,85,97] and other important Himalayan medicinal plants are projected to lose their suitable habitat under future climate scenarios [51,52,80].

Our results reveal that the decrease in the suitable habitats of *R. australe* is expected to be highest for the climate scenario: SSP585 for each of the four future periods (2021–2040,2041–2060,2061–2080, and 2081–2100) with the highest loss for the period 2081–2100 followed by the climate scenarios SSP370 and SSP 245 while the decrease in suitable habitat is expected to be the least for the climate scenario SSP126 (Table 3). Indeed, the losses are highly significant for two climate scenarios: SSP 585 and SSP 370 while nearly marginal for SSP 126 and SSP 245 (Tables 2 and 4). The emission scenario SSP 585 is characterized by higher radiative emission leading to a very high concentration of greenhouse gas, particularly carbon dioxide. Likewise, the emission scenario SSP 370 is also projected to possess a high concentration of carbon dioxide. In contrast, the two climate scenarios: SSP 126 and SSP 245 respectively represent the low radiation-low carbon dioxide emissions and the middle radiation-low carbon dioxide emission scenarios [54,55]. These results suggest that higher radiative emissions and higher concentrations of greenhouse gases are limiting the occurrence of *R. australe,* and thus it will face a significant reduction in suitable habitats with the intensification of atmospheric carbon dioxide concentration. Consistent with our findings, previous studies also reveal a decrease in suitable habitats with the rise of atmospheric carbon dioxide concentration [98,99]. The positive correlation between the extent of loss of suitable habitat of *R. australe* with the carbon dioxide emission severity could be attributed to low phenotypic plasticity as in several herbaceous weeds [99], decreased photosynthetic activity due to mechanical injury of stomata, or the consequences of weed invasion because increased carbon dioxide concentration favours the distribution, growth, and abundance of weeds [100,101]. Moreover, as in other mountainous regions also in the Nepalese Himalaya, the increased concentration of carbon dioxide is projected to create several indirect effects such as a rise in temperature, alteration in precipitation patterns, and change in vegetation phenology [10]. Shrestha et al. (2012) suggest that temperature (0.06 ˚C/year) and precipitation (6.52 mm/year) rise associated with climate change advance the average start and length of the growing season by 0.19 days per year while there is no impact on the end of the growing season in all the ecoregions of higher elevations in the Himalaya [10]. This change in phenology coupled with the rise in temperature and precipitation could likely negatively affect the distribution of *R. australe* in the Nepalese Himalaya. In addition, as in *R. webbianum*, the poor seed germination ability of *R. australe* under stressful environmental conditions may affect its propagation which eventually limits its distribution [80,97,102].

Our result reveals the overall contraction of suitable habitat of *R. australe* under all future climate scenarios, however, the findings are idiosyncratic at the individual district level. Three Districts: Humla, Darchula, and Bajhang represent the highest suitable areas for the current scenario but are expected to have the highest loss of suitable habitat in future climate scenarios. In contrast, Districts such as Jumla, Kalikot, Dolpa, Gorkha, and Jajarkot having relatively less suitable areas at the present condition are expected to have increased suitable areas for all (Jumla) or almost all future climate scenarios. This finding thus suggests the switch of suitable habitat of *R. australe* from the extremely dry higher western mountains towards relatively less dry lower mountains of Karnali regions and other similar areas. Consistent with our findings, previous findings also indicate the probable switch of suitable habitats for other medicinal plants towards the central mountains, the mountains that receive/ are expected to receive higher precipitation both for the current and future scenarios [23,51,52,103]. Our findings reveal that precipitation-related factors play a crucial role in *R. australe* distribution (Fig 3). Studies have suggested a downward trend both in the average annual rainfall and precipitation of the driest quarter in the

Table 4. The district-wise high suitable area (only including forests and rangelands above 2700m) (km²) of *Rheum australe* under different climate scenarios. Roman numbers I, II, III, and IV after socioeconomic pathways (SSPs) respectively represent the period for 2021-2040,2041-2060,2061-2080 and 2081-2100. The districts in the table are arranged in the order of high to low suitable areas of *R. australe* for the current scenario. Districts with boldface are expected to gain suitable habitats while districts with italicized are expected to lose suitable habitats under the future climate scenarios.

| Districts | Suitable area (km²) under different climate scenarios | | | | | | | | | | | | | | | | |
|---|---|---|---|---|---|---|---|---|---|---|---|---|---|---|---|---|---|
| | Current | ssp126-I | ssp126-II | ssp126-III | ssp126-IV | ssp245-I | ssp245-II | ssp245-III | ssp1245-IV | ssp370-I | ssp370-II | ssp370-III | ssp370-IV | ssp585-I | ssp585-II | ssp585-III | ssp585-IV |
| *Humla* | 136.92 | 78.46 | 63.66 | 71.79 | 39.97 | 66.61 | 53.30 | 57.00 | 57.76 | 74.77 | 70.34 | 33.32 | 28.13 | 78.47 | 54.78 | 36.28 | 11.85 |
| *Darchula* | 119.88 | 64.39 | 50.32 | 46.62 | 27.38 | 56.25 | 45.15 | 33.30 | 47.37 | 56.26 | 51.07 | 27.39 | 19.97 | 58.47 | 36.26 | 31.83 | 11.83 |
| *Bajhang* | 91.29 | 57.91 | 54.19 | 52.71 | 35.63 | 58.66 | 53.46 | 44.55 | 60.89 | 59.40 | 68.30 | 39.35 | 31.93 | 63.85 | 48.26 | 40.09 | 18.56 |
| **Jumla** | 61.96 | 104.52 | 98.53 | 93.32 | 100.78 | 94.06 | 128.40 | 139.59 | 138.85 | 156.01 | 154.51 | 128.4 | 101.53 | 130.63 | 137.35 | 138.10 | 76.15 |
| *Baglung* | 45.10 | 44.34 | 36.82 | 34.57 | 36.82 | 48.85 | 37.57 | 29.30 | 39.08 | 40.58 | 36.07 | 25.55 | 24.05 | 39.08 | 30.06 | 27.80 | 19.54 |
| *Myagdi* | 45.08 | 33.80 | 29.29 | 27.04 | 42.81 | 41.31 | 28.54 | 26.29 | 33.05 | 33.05 | 29.29 | 21.03 | 19.53 | 31.55 | 26.29 | 23.28 | 16.52 |
| *Sindhupal-chok* | 43.79 | 36.99 | 36.24 | 39.26 | 35.48 | 24.16 | 35.48 | 18.88 | 48.32 | 21.14 | 18.87 | 13.59 | 20.39 | 27.18 | 20.38 | 14.35 | 19.63 |
| **Kalikot** | 41.09 | 64.26 | 62.77 | 56.79 | 64.28 | 60.54 | 76.23 | 67.25 | 79.96 | 77.72 | 76.22 | 56.04 | 37.36 | 67.25 | 68.00 | 59.03 | 25.40 |
| **Dolpa** | 40.36 | 59.82 | 52.34 | 46.35 | 38.90 | 58.34 | 73.31 | 67.32 | 59.10 | 77.79 | 77.07 | 68.07 | 65.08 | 68.81 | 65.07 | 74.80 | 51.60 |
| *Solukhumbu* | 38.62 | 40.13 | 36.34 | 36.35 | 24.22 | 36.34 | 42.40 | 37.10 | 52.24 | 37.10 | 37.09 | 29.53 | 35.58 | 40.13 | 36.34 | 33.31 | 23.46 |
| *Sankhu-washabha* | 36.35 | 41.65 | 40.90 | 41.65 | 41.65 | 39.38 | 39.38 | 36.35 | 53.02 | 38.62 | 35.59 | 28.78 | 28.02 | 41.65 | 36.35 | 31.81 | 23.48 |
| *Dolakha* | 28.72 | 21.91 | 20.40 | 18.13 | 17.38 | 18.89 | 24.92 | 17.38 | 23.42 | 19.64 | 21.15 | 14.35 | 15.11 | 20.39 | 18.88 | 15.87 | 8.31 |
| *Rukum-East* | 28.52 | 44.28 | 32.26 | 29.26 | 30.76 | 48.03 | 39.77 | 27.76 | 36.02 | 45.77 | 36.76 | 25.51 | 21.01 | 40.52 | 27.76 | 26.26 | 16.51 |
| *Rasuwa* | 27.89 | 23.37 | 24.88 | 25.63 | 16.58 | 22.61 | 21.86 | 21.11 | 27.14 | 21.86 | 19.60 | 18.84 | 19.60 | 22.61 | 19.60 | 19.60 | 18.84 |
| *Taplejung* | 26.53 | 30.32 | 34.11 | 34.11 | 28.04 | 25.02 | 25.02 | 24.26 | 42.45 | 20.47 | 20.46 | 18.95 | 23.49 | 25.77 | 25.77 | 21.98 | 20.46 |
| *Bajura* | 23.77 | 27.53 | 20.84 | 14.88 | 19.36 | 29.02 | 35.73 | 30.52 | 37.97 | 32.74 | 37.97 | 24.56 | 11.91 | 32.75 | 32.01 | 25.30 | 5.95 |
| *Mugu* | 18.61 | 23.08 | 20.84 | 20.84 | 11.17 | 16.38 | 19.36 | 25.31 | 26.79 | 25.31 | 26.79 | 11.91 | 14.88 | 24.56 | 20.84 | 13.40 | 12.65 |
| **Gorkha** | 16.54 | 22.55 | 22.55 | 22.55 | 35.33 | 18.79 | 20.29 | 20.29 | 36.84 | 21.80 | 17.29 | 14.28 | 17.28 | 21.80 | 20.29 | 15.78 | 11.28 |
| *Kaski* | 12.79 | 9.03 | 8.27 | 8.27 | 9.77 | 9.03 | 8.27 | 8.27 | 9.03 | 8.27 | 8.27 | 7.52 | 5.27 | 8.27 | 8.27 | 8.27 | 4.51 |
| *Manang* | 12.77 | 18.03 | 16.52 | 15.02 | 27.79 | 17.27 | 15.77 | 12.77 | 18.78 | 19.53 | 11.27 | 7.51 | 5.26 | 17.27 | 13.52 | 9.01 | 7.51 |
| *Ramechhap* | 11.36 | 11.36 | 10.60 | 10.60 | 5.30 | 10.6 | 9.84 | 7.57 | 9.09 | 9.84 | 9.84 | 7.57 | 7.57 | 11.36 | 8.33 | 7.57 | 5.30 |
| *Lamjung* | 8.28 | 3.76 | 4.51 | 4.51 | 4.51 | 3.01 | 3.01 | 3.76 | 5.27 | 3.01 | 2.26 | 2.26 | 3.76 | 2.26 | 3.76 | 3.01 | 4.51 |
| *Mustang* | 6.00 | 5.25 | 4.50 | 4.50 | 16.50 | 5.25 | 5.25 | 5.25 | 6.75 | 8.25 | 4.50 | 3.75 | 6.00 | 6.75 | 5.25 | 3.75 | 4.50 |
| **Jajarkot** | 5.24 | 11.22 | 6.73 | 5.24 | 9.72 | 8.98 | 8.23 | 5.98 | 11.22 | 8.98 | 9.72 | 5.98 | 3.74 | 7.48 | 6.73 | 7.48 | 0.75 |
| *Rukum-West* | 2.25 | 3.75 | 3.00 | 3.00 | 0.75 | 3.75 | 3.00 | 1.50 | 3.00 | 3.00 | 3.00 | 1.50 | 1.50 | 3.00 | 1.50 | 1.50 | 0.75 |

**Table 5. Area of high suitable habitat (only including forest and rangeland) of *Rheum australe* under current and future climate scenarios in the entire country context.**

| Socio-economic Pathways (SSP)/ Time Periods | High suitable (only including forests and rangelands) area (km²) | High suitable area lying outside the current habitat type (km²) | Decrease in the high suitable area (km²) | The proportion of the decrease in the high suitable area (%) |
|---|---|---|---|---|
| Current (1970-2000) | 937.99 | 81.87 | | |
| ssp126: 2021–2040 | 889.96 | 60.56 | 48.03 | 5.12 |
| ssp126: 2041–2060 | 795.93 | 54.63 | 142.06 | 15.15 |
| ssp126: 2061–2080 | 767.51 | 51.64 | 170.48 | 18.18 |
| ssp126: 2081–2100 | 725.42 | 48.02 | 212.57 | 22.66 |
| ssp245: 2021–2040 | 829.38 | 49.33 | 108.61 | 11.58 |
| ssp245: 2041–2060 | 861.77 | 51.69 | 76.22 | 8.13 |
| ssp245: 2061–2080 | 773.17 | 45.71 | 164.83 | 17.57 |
| ssp245: 2081–2100 | 972.40 | 60.02 | -34.41 | -3.67 |
| ssp370: 2021–2040 | 926.14 | 58.28 | 11.85 | 1.26 |
| ssp370: 2041–2060 | 887.83 | 49.39 | 50.17 | 5.35 |
| ssp370: 2061–2080 | 637.04 | 44.95 | 300.95 | 32.08 |
| ssp370: 2081–2100 | 571.70 | 50.22 | 366.29 | 39.05 |
| ssp585: 2021–2040 | 897.11 | 56.81 | 40.88 | 4.36 |
| ssp585: 2041–2060 | 774.69 | 46.46 | 163.31 | 17.41 |
| ssp585: 2061–2080 | 615.95 | 75.79 | 246.26 | 26.25 |
| ssp585: 2081–2100 | 422.87 | 47.27 | 515.13 | 54.92 |

northwestern and western Himalaya [104–107]. Accordingly, these three districts being a part of the North-Western Himalaya are likely to receive decreased precipitation. Furthermore, the arid climatic conditions coupled with the complex land topography of these districts (located on the southern slope of Qinghai-Tibetan mountainous terrain) could result in decreased precipitation in the future. Moreover, it is likely that these three districts being the part/ located close to the Trans-Himalayan region would likely accumulate higher concentrations of greenhouse gases than other districts [9,68]. Thus, there could be a greater loss of suitable habitat for *R. australe* in these three districts.

*Rheum australe* is a widely used medicinal plant both in traditional and modern medical practices. Many rural communities across the country are engaged in the collection and trade of this plant which substantially contributes to their primary healthcare and livelihood [20]. In addition, the trade of this plant currently contributes 12.15% of the total national annual revenue from the medicinal plant trade, accounting for approximately 0.0021% of the national GDP [52]. Considering this significant trade potential, the government of Nepal has prioritized this plant as one of the 30 medicinal plants that could substantially contribute to the nation's economic prosperity [108]. This fact thus indicates that *R. australe* is one of the important national commodities connected not only with the primary healthcare and livelihood of local communities but also to the national economy. However, this important plant resource is encountering overharvest pressure in its wild populations, particularly due to its high demand in the medicinal market. Indeed, the overharvest pressure is already causing the depletion of this valuable resource from the wild [20] while unsustainable harvesting coupled with grazing and trampling has degraded several wild populations throughout the country (Paudel et al -unpublished data). As a result, *R. australe* has been the second most vulnerable traded MP of Nepal after *Nardostachys jatamansi* [20]. This threat is expected to be further exacerbated by future climate change as evidenced by our current findings which reveal the significant loss of its suitable habitat in all future climate scenarios. The loss of this important medicinal plant not only creates far-reaching

Table 6. The district-wise difference in high suitable area (only including forests and rangelands above 2700m) (km²) of *Rheum australe* under different climate scenarios with respect to the current suitable habitat. Roman numbers I, II, III, and IV after socioeconomic pathways (SSPs) respectively represent the time for 2021-2040,2041-2060,2061-2080 and 2081-2100. The districts in the table are arranged in the order of high to low projected loss of suitable habitat of *R. australe* for the climate scenario SSP126 for the period 2021-2040. Districts with boldface are expected to gain suitable habitats while districts with italicized are expected to lose suitable habitats under the future climate scenarios.

| Districts | Differences in Suitable area (km²) | | | | | | | | | | | | | | | |
|---|---|---|---|---|---|---|---|---|---|---|---|---|---|---|---|---|
| | ssp126-I | ssp126-II | ssp126-III | ssp126-IV | ssp245-I | ssp245-II | ssp245-III | ssp1245-IV | ssp370-I | ssp370-II | ssp370-III | ssp370-IV | ssp585-I | ssp585-II | ssp585-III | ssp585-IV |
| **Jumla** | 42.56 | 36.58 | 31.36 | 38.82 | 32.10 | 66.44 | 77.63 | 76.89 | 94.06 | 92.56 | 66.44 | 39.58 | 68.67 | 75.40 | 76.15 | 14.19 |
| **Kalikot** | 23.17 | 21.68 | 15.7 | 23.19 | 19.45 | 35.14 | 26.16 | 38.87 | 36.63 | 35.13 | 14.95 | -3.73 | 26.16 | 26.91 | 17.94 | -15.69 |
| **Dolpa** | 19.46 | 11.98 | 5.99 | -1.45 | 17.98 | 32.95 | 26.96 | 18.74 | 37.43 | 36.71 | 27.71 | 24.72 | 28.45 | 24.72 | 34.45 | 11.24 |
| Rukum-East | 15.75 | 3.74 | 0.74 | 2.24 | 19.50 | 11.24 | -0.76 | 7.49 | 17.25 | 8.24 | -3.01 | -7.51 | 11.99 | -0.76 | -2.26 | -12.02 |
| **Gorkha** | 6.01 | 6.01 | 6.01 | 18.79 | 2.25 | 3.75 | 3.75 | 20.30 | 5.25 | 0.74 | -2.27 | 0.74 | 5.25 | 3.75 | -0.76 | -5.27 |
| **Jajarkot** | 5.99 | 1.49 | 0.00 | 4.48 | 3.74 | 2.99 | 0.75 | 5.98 | 3.74 | 4.49 | 0.75 | -1.50 | 2.24 | 1.49 | 2.24 | -4.49 |
| Sankhu-washabha | 5.30 | 4.54 | 5.3 | 5.30 | 3.02 | 3.02 | 0.00 | 16.66 | 2.27 | -0.76 | -7.58 | -8.33 | 5.30 | 0.00 | -4.55 | -12.88 |
| Manang | 5.26 | 3.75 | 2.25 | 15.02 | 4.51 | 3.00 | 0.00 | 6.01 | 6.76 | -1.50 | -5.26 | -7.51 | 4.51 | 0.75 | -3.76 | -5.26 |
| Mugu | 4.47 | 2.23 | 2.23 | -7.44 | -2.23 | 0.75 | 6.70 | 8.18 | 6.70 | 8.18 | -6.70 | -3.73 | 5.95 | 2.23 | -5.21 | -5.96 |
| Taplejung | 3.79 | 7.58 | 7.58 | 1.51 | -1.52 | -1.52 | -2.28 | 15.91 | -6.07 | -6.07 | -7.58 | -3.04 | -0.76 | -0.76 | -4.55 | -6.07 |
| Bajura | 3.76 | -2.93 | -8.89 | -4.42 | 5.25 | 11.96 | 6.75 | 14.19 | 8.97 | 14.19 | 0.78 | -11.87 | 8.98 | 8.24 | 1.53 | -17.82 |
| Solukhumbu | 1.51 | -2.28 | -2.28 | -14.4 | -2.28 | 3.77 | -1.53 | 13.62 | -1.53 | -1.53 | -9.10 | -3.05 | 1.50 | -2.29 | -5.31 | -15.16 |
| Rukum-West | 1.50 | 0.75 | 0.75 | -1.50 | 1.50 | 0.75 | -0.75 | 0.75 | 0.75 | 0.75 | -0.75 | -0.75 | 0.75 | -0.75 | -0.75 | -1.50 |
| Ramechhap | 0.00 | -0.76 | -0.76 | -6.06 | -0.76 | -1.52 | -3.79 | -2.27 | -1.52 | -1.52 | -3.79 | -3.79 | 0.00 | -3.03 | -3.79 | -6.06 |
| Mustang | -0.75 | -1.50 | -1.50 | 10.50 | -0.75 | -0.75 | -0.75 | 0.75 | 2.25 | -1.50 | -2.25 | 0.00 | 0.75 | -0.75 | -2.25 | -1.50 |
| *Baglung* | -0.76 | -8.27 | -10.53 | -8.28 | 3.75 | -7.52 | -15.79 | -6.02 | -4.52 | -9.03 | -19.55 | -21.05 | -6.02 | -15.04 | -17.29 | -25.56 |
| *Kaski* | -3.76 | -4.51 | -4.51 | -3.01 | -3.76 | -4.51 | -4.51 | -3.76 | -4.51 | -4.51 | -5.27 | -7.52 | -4.51 | -4.51 | -4.51 | -8.27 |
| *Lamjung* | -4.52 | -3.76 | -3.76 | -3.76 | -5.27 | -5.27 | -4.51 | -3.01 | -5.27 | -6.02 | -6.02 | -4.51 | -6.02 | -4.51 | -5.27 | -3.76 |
| *Rasuwa* | -4.52 | -3.02 | -2.26 | -11.31 | -5.28 | -6.03 | -6.79 | -0.75 | -6.03 | -8.29 | -9.05 | -8.29 | -5.28 | -8.29 | -8.29 | -9.05 |
| *Sindhupal-chok* | -6.80 | -7.56 | -4.54 | -8.32 | -19.63 | -8.31 | -24.92 | 4.52 | -22.66 | -24.92 | -30.20 | -23.41 | -16.62 | -23.41 | -29.45 | -24.17 |
| *Dolakha* | -6.81 | -8.32 | -10.59 | -11.34 | -9.83 | -3.79 | -11.34 | -5.30 | -9.08 | -7.57 | -14.36 | -13.61 | -8.32 | -9.83 | -12.85 | -20.41 |
| *Myagdi* | -11.28 | -15.79 | -18.04 | -2.27 | -3.76 | -16.54 | -18.79 | -12.03 | -12.03 | -15.79 | -24.05 | -25.55 | -13.53 | -18.79 | -21.80 | -28.56 |
| *Bajhang* | -33.38 | -37.09 | -38.58 | -55.66 | -32.63 | -37.82 | -46.73 | -30.40 | -31.89 | -22.98 | -51.94 | -59.36 | -27.44 | -43.03 | -51.20 | -72.72 |
| *Darchula* | -55.49 | -69.56 | -73.26 | -92.5 | -63.62 | -74.73 | -86.57 | -72.50 | -63.62 | -68.80 | -92.49 | -99.91 | -61.40 | -83.61 | -88.05 | -108.05 |
| *Humla* | -58.46 | -73.27 | -65.13 | -96.96 | -70.31 | -83.63 | -79.92 | -79.17 | -62.16 | -66.59 | -103.60 | -108.8 | -58.46 | -82.14 | -100.64 | -125.08 |

consequences on the traditional medicinal practice and livelihood of many rural communities but also affects the national economy. Moreover, given that *R. webbianum* serves as an important food resource host for the keystone wild pollinators such as bumblebees (*Bombus* spp.), carpenter bees (*Xylocopa* spp.), honeybees (*Apis* spp.), sweat bees (*Lasioglossum* spp.), and wasps (*Aphelinu*s spp.) [109], the loss of this plant species likely has a direct impact on the survival of many important wild pollinators which eventually affects the pollination network and the community composition. Nonetheless, we yet lack intensive conservation efforts for this valuable plant resource. These facts thus underlie the need to devise effective strategies for its conservation and sustainable utilization.

Given the uncertainty of future climate scenarios and great differences in the projected atmospheric carbon dioxide concentration among the various projections, accurately predicting the impact of climate change on the distribution of plant species and devising effective strategies for their conservation is quite challenging. Our current finding comprises the habitat prediction of *R. australe* under all future possible scenarios, and thus it provides meaningful insight to devise emission scenario-specific conservation strategies. The prediction models reveal that under the emission scenarios: SSP 126 and SSP 245, most of the current suitable habitats are predicted to be retained while for SSP 370 and SSP 585, significant loss is expected. Therefore, we suggest that under SSP 126 and SSP 245, tracking and monitoring of suitable habitats to reduce anthropogenic disturbances would be effective in the conservation of *R. australe*. In contrast, for scenarios SSP 370 and SSP 585, sites-specific integrated conservation approaches such as monitoring of suitable habitats to reduce anthropogenic disturbances, habitat restoration, and reintroduction (assisted colonization/ translocation) would be required.

Extending the current finding to the local levels, the result reveals that districts from the lower mountainous regions such as Jumla, Kalikot, Dolpa, Jajarkot, and Gorkha will have increased suitable areas for almost all future climate scenarios (Table 5). Therefore, the conservation of *R. australe* in these districts could be achieved simply by tracking and monitoring the natural habitats. In contrast, three districts from the North-Western Himalaya (Humla, Darchula, and Bajhang) having the highest suitable areas at the present condition are expected to have a huge loss of suitable habitats (up to 91.34% loss in Humla in the scenario SSP585 for the period 2081–2100) in all future climate scenarios. Likewise, Districts: Myagdi, Dolakha, Sindhupalchok, Rasuwa, Lamjung, Kaski, and Baglung are predicted to lose suitable habitats under all future climate scenarios. The significant reduction of suitable habitat of *R. australe* may increase their vulnerability to other threats, and thus specific conservation strategies are required for these districts. Based on the extent of habitat loss, we suggest that habitat restoration would help the conservation of *R. australe* in the districts: Myagdi, Dolakha, Sindhupalchok, Rasuwa, Lamjung, Kaski, and Baglung while short distance assisted colonization within the protected area system would be required as a proactive conservation strategy of *R. australe* in the districts: Humla, Darchula, and Bajhang. Moreover, considering the findings that the Jumla district will have significantly increased suitable habitats for all future scenarios, and other districts such as Kalikot, Dolpa, Jajarkot, and Gorkha will have increased suitable areas for almost all the future climate scenarios (Table 5), we suggest that forests and rangelands of these districts lying at an elevation range of 3300 m – 4,400 m could be developed as potential planting areas for commercial cultivation as well as for rewilding, restoration and establishing germplasm conservation centres. On the other hand, the large area of the current suitable habitat from the districts Humla, Darchula, Bajhang, Myagdi, Dolakha, Sindhupalchok, Rasuwa, Lamjung, Kaski, and Baglung will become unsuitable under all future climate scenarios. Therefore, these districts are deemed unsuitable for the domestication of germplasm conservation centres and hence proper attention should be paid if planting areas/conservation sites are to be developed within these districts. This finding thus has wider policy implications for both government and conservation organizations, and medicinal plants farmers at large.

## Supporting information

**S1 Table.  Cleaned and Rarefied occurrence data of *Rheum australe* in Nepal.** These occurrence points were used to construct the distribution model of *R. australe* for the current and different future scenarios.
(XLSX)

## Acknowledgments

We thank the Department of Forest, the Department of National Parks and Wildlife Conservation, and the Department of Plant Resources, Government of Nepal for providing research permission. We acknowledge the help provided by Mr. Shiva Pokhrel in analyzing the spatial data. We thank Dr Subodh Adhikari for his valuable input in revising the manuscript.

## Author contributions

**Conceptualization:** Babu Ram Paudel, Chandra Kanta Subedi, Dipesh Pyakurel, Meena Rajbhandari, Ram Prasad Chaudhary.

**Data curation:** Babu Ram Paudel, Suresh Kumar Ghimire.

**Formal analysis:** Babu Ram Paudel.

**Funding acquisition:** Babu Ram Paudel, Dipesh Pyakurel, Meena Rajbhandari, Ram Prasad Chaudhary.

**Investigation:** Babu Ram Paudel.

**Methodology:** Babu Ram Paudel, Dipesh Pyakurel.

**Project administration:** Babu Ram Paudel, Chandra Kanta Subedi, Ram Prasad Chaudhary.

**Resources:** Babu Ram Paudel, Suresh Kumar Ghimire.

**Software:** Babu Ram Paudel.

**Supervision:** Babu Ram Paudel, Ram Prasad Chaudhary.

**Validation:** Babu Ram Paudel.

**Visualization:** Babu Ram Paudel.

**Writing – original draft:** Babu Ram Paudel.

**Writing – review & editing:** Babu Ram Paudel, Chandra Kanta Subedi, Suresh Kumar Ghimire, Dipesh Pyakurel, Meena Rajbhandari, Ram Prasad Chaudhary.

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
