## [Decision Letter · Decision Letter 0]

17 Dec 2024

PONE-D-24-47340Impacts of climate change on the distribution pattern of Himalayan Rhubarb (Rheum australe D.Don) in Nepal HimalayaPLOS ONE

Dear Dr. Paudel,

Thank you for submitting your manuscript to PLOS ONE. After careful consideration, we feel that it has merit but does not fully meet PLOS ONE’s publication criteria as it currently stands. Therefore, we invite you to submit a revised version of the manuscript that addresses the points raised during the review process.

We look forward to receiving your revised manuscript.

Kind regards,

Daniel de Paiva Silva, Ph.D.

Academic Editor

PLOS ONE

Journal Requirements:

“The study was supported by an innovative research grant (Grant Number: TU-NPAR-078/79-4-04) from the Research Directorate, Office of the Rector, Tribhuvan University, Nepal.”

3. We note that Figures 1, 4, 5 and 6 in your submission contain [map/satellite] images which may be copyrighted. All PLOS content is published under the Creative Commons Attribution License (CC BY 4.0), which means that the manuscript, images, and Supporting Information files will be freely available online, and any third party is permitted to access, download, copy, distribute, and use these materials in any way, even commercially, with proper attribution. For these reasons, we cannot publish previously copyrighted maps or satellite images created using proprietary data, such as Google software (Google Maps, Street View, and Earth). For more information, see our copyright guidelines: http://journals.plos.org/plosone/s/licenses-and-copyright .

     1. You may seek permission from the original copyright holder of Figures 2, 3, 5 and 6  to publish the content specifically under the CC BY 4.0 license. 

Additional Editor Comments:

Dear Dr. Paudel,

After this first review round you have received a major and a minor reviews. THerefore, you manuscript may be accepted for publication in PLoS One after a major review is considered. Please take careful consideration on the suggestions provided by both reviewers. Considering the extent of reviews you need to do, I believe you will need a three-month period to complete the changes required by the reviewers. Please do not forget to prepare a rebuttal letter, showing the changes you performed following the reviewers' suggestions and those that you did not agree with, informing your reviewers the reason why the changes were not performed.

Sincerely,

Daniel Silva

Associate Editor

Reviewers' comments:

Reviewer's Responses to Questions

**Comments to the Author**

1. Is the manuscript technically sound, and do the data support the conclusions?

Reviewer #1: Partly

Reviewer #2: Yes

2. Has the statistical analysis been performed appropriately and rigorously? 

Reviewer #1: Yes

Reviewer #2: Yes

3. Have the authors made all data underlying the findings in their manuscript fully available?

Reviewer #1: Yes

Reviewer #2: Yes

4. Is the manuscript presented in an intelligible fashion and written in standard English?

Reviewer #1: Yes

Reviewer #2: Yes

5. Review Comments to the Author

Reviewer #1: Here are my suggestions: Depth in Discussion of Climate Impacts: While mentioning the general effects of climate change, the introduction could explore in more detail how these impacts affect Nepal in particular (would CO2 emissions influence plant distribution/suitability??? Would rainfall and temperature alter photosynthetic rate???), with an emphasis on interactions between environmental and socioeconomic factors.... It is interesting to clarify what the effect of climate change is.

Part of the introduction could be moved to the discussion, as it contains a critical analysis of previous studies, methodological limitations, and implications for future research—typical elements of a discussion section. Specifically, the following points would be more appropriate in the discussion: 1. **Model and Data Comparison**: An assessment of the limitations of CMIP5 data and a rationale for using CMIP6 data, including specific benefits in terms of reduced lifespans and improved predictions in climate forecasts (lines 85–94). And lines 95-108... This excerpt is a critical evaluation of previous studies, highlighting limitations and justifying the methodological and conceptual advances of the current study. As such, it would be more appropriate in the discussion section of the article, preferably in a paragraph that analyzes how the results found compare with the existing literature.

Lines 60-83, in the excerpt provided, the author does not clearly specify whether he used SDM (Species Distribution Modeling) or ENM (Ecological Niche Modeling) in the study. He mentions both terms interchangeably, but the focus seems to be on modeling the distribution of the species based on occurrence data and environmental variables, which is characteristic of both SDM and ENM.

 ...MaxEnt is often associated with species distribution modeling (SDM). However, the methodology can also be considered within the context of ENM, since it involves the use of environmental variables to estimate the ecological niche of the species. Therefore, the author does not make an explicit distinction... Make it clear to the reader what was done.

In the Natural Distribution Data Section: mention the date of access to the databases.

Environmental Variables: it is necessary to justify the choice for the use of each variable. The reader needs to understand the role of temperature and precipitation in the species.

Why was the variable solar radiation not considered? Solar radiation is essential for plant growth and development, as it is the source of energy for photosynthesis.

Results: Improve the explanation of the criteria used for the classification of habitat zones and the organization of data in the tables.

The figures and graphs mentioned in the article (such as Figures 4-6) are essential to illustrate the projections, but the details on how each figure contributes to the understanding of the results could be more explanatory. This includes the description of what each colored area represents and how it relates to the zones of suitable habitat in each scenario.

Appreciation of AUC and TSS Values: Although the AUC (0.921) and TSS (0.68) values are mentioned as indicators of model accuracy, a clearer interpretation of these numbers would be useful. For example, it could be detailed what these results imply for the reliability of the projections in different climate scenarios. It would also be interesting to discuss the impact of a TSS of 0.68 in terms of model predictability and compare it with other similar studies.

In the discussion:

I suggest removing the subtitles, the discussion section becomes more fluid and less segmented, which makes it easier to understand the text as a whole.

Deeper exploration of limitations: Although the discussion mentions methodological advances, a more detailed analysis of the limitations of the study could be useful. For example, the accuracy of species occurrence data and the challenges associated with the use of climate models (such as uncertainty in climate scenarios) could be better explored. Expanding on comparisons with other studies: Although the discussion briefly mentions the contributions of the study to the existing literature, it would be interesting to see a more in-depth comparison with other similar studies on the distribution of medicinal plants in mountainous regions or the impact of climate change on other species. This would help to better contextualize the findings.

Clarity on the implications of the results for the future: The text could provide more detail on how the results may impact the management and conservation of R. australe in the long term. It would be interesting to expand on the policy recommendations, considering different climate change scenarios and how they could affect cultivation and conservation areas.

Clarity on the implications of the results for the future: The text could provide more detail on how the results may impact the management and conservation of R. australe in the long term. It would be interesting to expand the policy recommendations, considering different climate change scenarios and how they could affect cultivation and conservation areas.

Reviewer #2: Thank you for the opportunity to learn from your work.

This is an interesting paper that contributes to the debate about the effects of climate change over economically valuable species. However, the Discussion needs to be revised to make sense in a journal like Plos One. The results will be better explored if the ecological and economic impacts of the extinction of Rheum australe are addressed in greater depth and in a broader way, moving away from the logic of dialogue between similar articles.

I'm sending my considerations line by line below.

Line 30: Those keywords are already on title; please replace them.

Line 83: It is also important to talk about the limitations of the method (Maxent) somewhere in the text.

Line 147: Fig 1. Please, improve the plant image resolution.

Line 284: Please italicize “Rheum australe”.

Line 296: Consider excluding terms that can be seen as judgment value, as “our ’extensive’ observation”, “actual high”.

Lines 307-310: But wouldn't it be possible for this environment to also be suitable? What if the species just didn't make it there? My point is, given that the model indicates these areas as suitable, and how severe climate change could be, aren't you being too conservative in considering only forests and ragelands as suitable areas? I'd like to hear your thoughts on this.

Lines 341-342: I couldn’t find the source of Land Use and Land Cover result, once it is not in Jackknife analysis or elsewhere. Please, make it clearer. Also, there was not % residual?

Table 2: The sum of Contribution) is 100.1, and Permutation importance is 100.2. Please, re-round.

Line 366: Figure 3. Please, indicate the figure as 3a, 3b, 3c... in the figure and in the manuscript. It was a waste of time to connect figures and the correspondent text.

Line 374: Put you graphic in %, following the manuscript.

Line 375: You forgot to mention the temperature in the driest quarter.

Line 380: “Probability of presence”: Considering the statistical meaning of "probability", is it correct in a strict sense? If you are not sure about this, please replace it.

Lines 388 and 389: Be nice to you reader and enter the data here (%).

Line 392: Potential or current?

Lines 446 and 448: Could you please mark those cases on the table? Also, standardize the decimal places, and double-check.

Line 451: Table 4: Same as Line 296.

Line 460: Under 15 what?

Line 492: Aspect: I suggest changing this term, since "aspect" can mean anything when alone.

Lines 506 and 507: It's okay to use Maxent, you don't have to justify yourself too much. But it's important to consider and make clear to the reader the limitations of this choice.

Lines 509 and 511: Double-check the sentence, it seems weird.

Line 535: You spent a lot of time comparing your work with the work of other authors who talk about species other than yours. It seems quite clear to me that each species will respond differently to environmental variables. I suggest you rethink your approach. Suggestion: Explore the fact that environmental factors, such as elevation (not temperature), are so important to the species. What does this tell us?

Line 538: You say "studies", but you only point to one reference.

Lines 541-542: Why?

Lines 552-554: So... ? Why this is important? Did they use other approach, for example?

Line 556: The fact that you have a general shrinkage of the suitable area and an increase in these areas in certain locations, does not mean that your results are inconsistent. Please choose a better word.

Lines 561-563: This is a nice result!!

Line 570: And what does your models say?

Line 578: replace “;” to “,”.

Line 582: What characterizes these scenarios?

Lines 583-584: Are high emissions or their consequences limiting the occurrence of the species?

Lines 587-592: But here, in the Discussion section, you must discuss this result, not just let it go until another study comes along.

Lines 598-599: You should discuss this, not just say you don't know. It's okay to recognize the limitations of the model, but that doesn't prevent you from discussing the results based on the SDM science that exists to date.

Lines 619-621: Please, rewrite.

Line 634: Here you are making recommendations based on your findings. It is fine, but now you see how this recommendations are incongruous if you think about you tell us in the last section?

6. PLOS authors have the option to publish the peer review history of their article (what does this mean? ). If published, this will include your full peer review and any attached files.

**Do you want your identity to be public for this peer review?** For information about this choice, including consent withdrawal, please see our Privacy Policy .

Reviewer #1: No

Reviewer #2: No

---

## [Author Response · Author response to Decision Letter 1]

18 Jan 2025

Dear Daniel de Paiva Silva, Ph.D.

Academic Editor

PLOS ONE,

Thank you for providing an opportunity to revise the manuscript “PONE-D-24-47340 Impacts of climate change on the distribution pattern of Himalayan Rhubarb (Rheum australe D.Don) in Nepal Himalaya”.We have thoroughly revised the manuscript based on the suggestions/comments of the reviewers. Indeed, the comments/suggestions given by the reviewers helped us immensely to improve the clarity of the manuscript. We provide the point-by-point responses to the editorial and reviewers’ comments as an attachment in the submission system as the file name"Resaponse to Reviewers". We now believe that the revised manuscript is suitable for publication in PLOSONE.

Sincerely yours

Babu Ram Paudel

---

## [Decision Letter · Decision Letter 1]

15 Apr 2025

Impacts of climate change on the distribution pattern of Himalayan Rhubarb (Rheum australe D.Don) in Nepal Himalaya

PONE-D-24-47340R1

Dear Dr. Paudel,

We’re pleased to inform you that your manuscript has been judged scientifically suitable for publication and will be formally accepted for publication once it meets all outstanding technical requirements.

Kind regards,

Daniel de Paiva Silva, Ph.D.

Academic Editor

PLOS ONE

Additional Editor Comments (optional):

Reviewers' comments:

Reviewer's Responses to Questions

**Comments to the Author**

1. If the authors have adequately addressed your comments raised in a previous round of review and you feel that this manuscript is now acceptable for publication, you may indicate that here to bypass the “Comments to the Author” section, enter your conflict of interest statement in the “Confidential to Editor” section, and submit your "Accept" recommendation.

Reviewer #2: All comments have been addressed

2. Is the manuscript technically sound, and do the data support the conclusions?

Reviewer #2: Yes

3. Has the statistical analysis been performed appropriately and rigorously? 

Reviewer #2: Yes

4. Have the authors made all data underlying the findings in their manuscript fully available?

Reviewer #2: Yes

5. Is the manuscript presented in an intelligible fashion and written in standard English?

Reviewer #2: Yes

6. Review Comments to the Author

Reviewer #2: Now that all the comments have been properly addressed, I believe the paper is ready to be submitted for publication in PLOS ONE.

7. PLOS authors have the option to publish the peer review history of their article (what does this mean? ). If published, this will include your full peer review and any attached files.

**Do you want your identity to be public for this peer review?** For information about this choice, including consent withdrawal, please see our Privacy Policy .

Reviewer #2: No

---

## [Editor Report · Acceptance letter]

PONE-D-24-47340R1

PLOS ONE

Dear Dr. Paudel,

I'm pleased to inform you that your manuscript has been deemed suitable for publication in PLOS ONE. Congratulations! Your manuscript is now being handed over to our production team.

Kind regards,

on behalf of

Dr. Daniel de Paiva Silva

Academic Editor

PLOS ONE